# DecoderTCR: Compositional Pretraining and
# Entropy-Guided Decoding for TCR-pMHC Interactions

**Boqiao Lai** [1]   **Melissa Englund** [1]   **Ramit Bharanikumar** [1]   **Isabel Nocedal** [1]   **Ali Davariashtiyani** [1 2]
**Jason Perera** [1]   **Aly A. Khan** [1 2]

## Abstract

Modeling recognition between T-cell receptors (TCRs) and peptide-MHC (pMHC) complexes is a fundamental challenge in computational immunology, constrained by sparse paired interaction data relative to abundant unpaired sequences. We introduce **DecoderTCR**, a masked language model framework that addresses this through two contributions: (1) a **compositional continual pre-training curriculum** that learns component representations from marginal data before refining cross-chain dependencies, and (2) **Iterative Entropy-Guided Refinement (IEGR)**, a non-autoregressive decoding algorithm that resolves high-confidence positions first to provide context for uncertain regions. On held-out benchmarks, DecoderTCR achieves 0.96 AUROC for zero-shot pMHC binding prediction and 0.76 AUROC for epitope-specific TCR recognition, approaching supervised baselines without epitope-specific training. Learned representations recover structural contacts without coordinate supervision, and generated sequences exhibit realistic recombination statistics. Experimental validation across two rounds of wet-lab screening reveals a prediction-generation gap that can be narrowed via a lab-in-the-loop paradigm for TCR design.

## 1. Introduction

The specificity of adaptive immunity is determined by molecular recognition between T-cell receptors (TCRs) and peptide-major histocompatibility complexes (pMHCs). This interaction underlies protective responses to infection and cancer, while off-target recognition triggers autoimmunity.

Controlling this mechanism provides the basis for T-cell therapies that redirect immune recognition toward disease-associated antigens. A central goal in computational immunology is to predict and design TCRs that recognize a desired pMHC target with high specificity.[1]

Protein language models (pLMs) trained on large corpora learn structural and functional regularities from sequence alone (Lin et al., 2023). However, applying generic pLMs to immune recognition faces two challenges. First, TCR-pMHC recognition is mediated by a multi-component interface involving TCR $\alpha/\beta$ chains, peptide, and MHC complex. Models must capture cross-chain dependencies rather than marginal constraints within individual sequences. Second, generative design for such interfaces is intrinsically conditional and localized, requiring design of specific regions such as CDR loops while conditioning on a fixed antigenic context.

These challenges are compounded by severe data heterogeneity: TCR repertoire studies yield $\sim 10^7$ TCR sequences and mass spectrometry provides $\sim 10^6$ pMHC ligandome interactions, yet paired interaction data remains orders of magnitude smaller ($< 10^5$), noisy, and biased toward common alleles and peptides (Figure 1A). This motivates use of a multi-stage continual pretraining framework (Ke et al., 2023; Gururangan et al., 2020) that leverages component-level representation learning from abundant marginal data before refining cross-component dependencies from limited paired interactions.

Prior TCR-pMHC discriminative methods learn binding classifiers (Montemurro et al., 2021; Lu et al., 2021; Weber et al., 2021) but cannot directly generate sequences. Autoregressive protein language models (Madani et al., 2023; Ferruz et al., 2022) can sample novel sequences, but left-to-right factorization is poorly suited for constrained TCR loop redesign where residues are jointly constrained by both N-terminal and C-terminal context. Structure-conditioned methods (Dauparas et al., 2022; Watson et al., 2023) require coordinate supervision which remains scarce for TCR-pMHC interfaces ($< 10^3$ solved structures).

---

[1]Biohub, Chicago, IL, USA [2]University of Chicago, Chicago, IL, USA. Correspondence to: Aly A. Khan <aakhan@uchicago.edu>.

*Proceedings of the 43$^{rd}$ International Conference on Machine Learning*, Seoul, South Korea. PMLR 306, 2026. Copyright 2026 by the author(s).

---

[1]Appendix A briefly reviews TCR-pMHC biology.

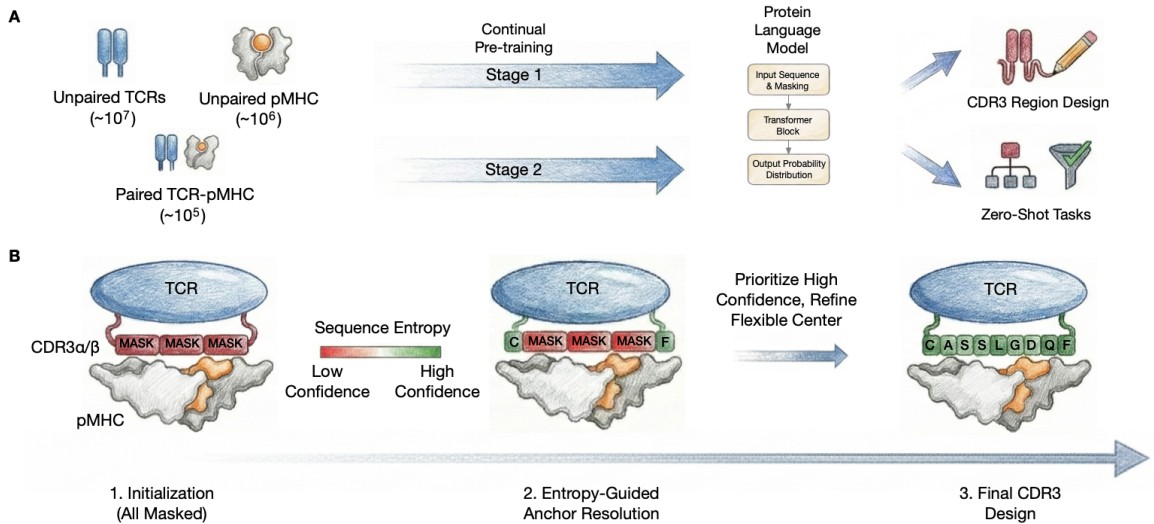

*Figure 1.* **Overview of DecoderTCR. (A)** Component-specific masked modeling integrates abundant unpaired TCR repertoires ($\sim 10^7$) and pMHC data ($\sim 10^6$) with sparse paired TCR-pMHC interactions ($< 10^5$) via tailored masking rates. **(B)** Iterative Entropy-Guided Refinement (IEGR) performs constrained CDR3 redesign by resolving low-entropy anchor positions first, followed by local masked resampling of flexible regions.

We introduce **DecoderTCR**, a sequence-first framework addressing TCR-pMHC *prediction* and *design* through two innovations (Figure 1). First, we propose a **multi-stage continual pre-training curriculum** that bridges heterogeneous data sources. Rather than learning the joint distribution $p(\mathbf{y}, \mathbf{x})$ from sparse paired data alone, we continually pre-train ESM-2 with component-specific masking schedules tailored to each immune component. This strategy allows abundant marginal data to regularize learning while limited paired data refines cross-chain dependencies.

Second, we propose **Iterative Entropy-Guided Refinement (IEGR)**, a non-autoregressive decoding strategy for constrained CDR3 redesign. Building on confidence-based iterative decoding (Chang et al., 2022; Ghazvininejad et al., 2019), IEGR iteratively unmasks positions in order of predictive entropy, resolving low-entropy anchor residues first to establish a stable scaffold, then refining high-entropy flexible regions through local masked resampling. This schedule respects the biophysical hierarchy where conserved positions constrain hypervariable loops.

We evaluate DecoderTCR across representation quality, zero-shot binding prediction, and experimental validation. On held-out benchmarks with stringent allele and epitope splits, DecoderTCR achieves 0.96 AUROC on zero-shot pMHC binding (compared to 0.45 for ESM-2) and 0.76 AUROC on epitope-specific TCR recognition, approaching supervised baselines trained explicitly on target epitopes. Peptide interaction scores localize structural contact positions despite training without coordinate supervision. Experimental screening of designed TCRs confirms that zero-shot generation remains challenging.

Our contributions are:

1. **Compositional continual pre-training.** A two-stage curriculum bridging the $100\times$ gap between marginal sequences ($\sim 10^7$ TCRs, $\sim 10^6$ pMHCs) and paired interactions ($\sim 10^5$) via functionally-informed masking and implicit experience replay. To our knowledge this is the first multi-component pLM that performs *joint* modeling over the full immune synapse for both TCR-pMHC prediction and zero-shot design.

2. **Entropy-guided decoding with span scoring.** Iterative Entropy-Guided Refinement (IEGR) resolves low-entropy anchors before high-entropy hypervariable regions, producing sequences with realistic recombination statistics and structural plausibility. Span pseudo-likelihood scoring enables binding prediction and recovery of structural contacts without coordinate supervision.

3. **Comprehensive evaluation.** Zero-shot generalization on strictly held-out alleles and epitopes, two rounds of wet-lab validation demonstrating a lab-in-the-loop design paradigm ($3\% \rightarrow 12\%$ hit rate), cross-paradigm comparison against structure-based methods, and rigorous characterization of the prediction-generation gap.

## 2. Related Work

**Protein Language Models for Immune Repertoires.** Large-scale protein language models learn structural and functional constraints from sequence alone (Lin et al., 2023; Elnaggar et al., 2021). Immune-focused models including TCR-BERT (Wu et al., 2024), TCRLM (Fang et al.,

2024), ESMCBA (Mares et al., 2025), and IgLM (Shuai et al., 2023) support repertoire-level tasks, such as clonotype clustering and motif discovery. These models capture marginal regularities within single sequence families rather than conditional dependencies across multicomponent receptor-antigen interfaces. Structure-enhanced PLMs (ESM3 (Hayes et al., 2025), ProSST (Li et al., 2024), DPLM (Wang et al., 2024)) augment pretraining with geometric features, but fewer than 500 solved TCR-pMHC ternary structures exist, leaving such signal sparse. Our component-specific masking strategy combines marginal priors from abundant unpaired data with cross-chain dependencies from limited paired interactions.

**TCR-pMHC Binding Prediction.** For pMHC binding, transformer-based methods such as TransPHLA (Chu et al., 2022) and HPL (Zhu et al., 2025) complement allele-specific predictors (O'Donnell et al., 2020; Reynisson et al., 2020; Gfeller et al., 2023). For TCR-pMHC recognition, deep models including NetTCR-2.0 (Montemurro et al., 2021), ERGO-II (Springer et al., 2021), pMTnet (Lu et al., 2021), TITAN (Weber et al., 2021), ATM-TCR (Cai et al., 2022), STAPLER (Kwee et al., 2023), and DAISY (Yuan et al., 2026) have advanced beyond similarity kernels (Dash et al., 2017). Springer et al. (2021) showed that including both CDR3$\alpha$ and CDR3$\beta$ substantially improves binding accuracy, a design choice we adopt throughout. The IMM-REP23/25 benchmarks (Nielsen et al., 2024; Richardson et al., 2026) document that these supervised methods degrade on unseen epitopes (Lu et al., 2026), and none directly generate sequences. DecoderTCR addresses both limitations: it jointly models both CDR3 chains and enables generative sampling while leveraging unpaired data to improve out-of-distribution generalization.

*De Novo* **TCR Design.** Sequence-based generative methods (ERTransformer (Yang et al., 2023), TCR-TRANSLATE (Karthikeyan et al., 2025)) produce candidate CDRs but operate on a single chain, use autoregressive decoding, or condition only on peptide rather than the full multi-component interface. Structure-based pipelines (RFdiffusion+ProteinMPNN (Watson et al., 2023; Dauparas et al., 2022), BoltzGen (Stark et al., 2025), and the TCR-specific ADAPT (Motmaen et al., 2025)) require structural templates and do not attempt zero-shot generalization to targets lacking solved or co-folded structures, motivating sequence-first methods that operate in the template-scarce regime characteristic of clinically relevant antigens. AlphaFold3 (Abramson et al., 2024) is a structure *predictor* rather than a sequence generator, and we use it as an *in silico* oracle (Section 5.5). Appendix I contrasts these paradigms.

**Continual Pre-training and Non-Autoregressive Decoding.** Continual pre-training adapts foundation models to specialized domains (Gururangan et al., 2020), with curriculum strategies mitigating catastrophic forgetting under distribution shift (Ke et al., 2023). Our two-stage curriculum extends this to multi-component protein interfaces where heterogeneous data sources require distinct masking strategies. For generation, Mask-Predict (Ghazvininejad et al., 2019) and MaskGIT (Chang et al., 2022) introduced iterative masked decoding with confidence-based token selection. Wang & Cho (2019) treat masked-LM conditionals as a Gibbs sampler. These iterative-decoding methods can also be viewed as absorbing-state discrete diffusion samplers (Austin et al., 2021), with masking corresponding to the forward noising process. The three strategies we evaluate (OneShot, Greedy, IEGR) instantiate single-pass parallel sampling, argmax decoding, and confidence-ordered iterative unmasking with block Gibbs refinement (recovering Wang & Cho (2019) at block size $b{=}1$), adapted to constrained protein design over a fixed-length scaffold.

## 3. Methods

We formulate TCR-pMHC modeling as learning conditional distributions over the full multi-component immune synapse. Our approach comprises four components: a unified sequence representation, component-aware masked language modeling, span pseudo-likelihood scoring for binding prediction, and entropy-guided decoding for generation.

**Notation.** Let $\mathcal{V}$ denote the amino acid vocabulary with $|\mathcal{V}| = 20$ canonical residues. We denote pMHC sequences as $\mathbf{x} = (x_1, \ldots, x_n) \in \mathcal{V}^n$ and TCR sequences as $\mathbf{y} = (y_1, \ldots, y_m) \in \mathcal{V}^m$. The joint TCR-pMHC complex is represented as concatenation $\mathbf{z} = [\mathbf{x} \,\|\, \mathbf{y}] \in \mathcal{V}^{n+m}$. For sequence $\mathbf{s} = (s_1, \ldots, s_L)$ and position set $\mathcal{M} \subseteq [L] := \{1, \ldots, L\}$, we write $\tilde{\mathbf{s}}^{(\mathcal{M})}$ for the sequence with all positions in $\mathcal{M}$ replaced by [MASK].

### 3.1. Unified Multi-Chain Representation

To capture cross-chain dependencies, we represent the TCR-pMHC complex as a single concatenated sequence:

$$\mathbf{z} = \big[\underbrace{\mathbf{s}_{\mathrm{MHC}\alpha} \,\|\, \mathbf{s}_{\mathrm{MHC}\beta} \,\|\, \mathbf{s}_{\mathrm{pep}}}_{\mathbf{x}} \,\|\, \underbrace{\mathbf{s}_{\alpha} \,\|\, \mathbf{s}_{\beta}}_{\mathbf{y}}\big], \quad (1)$$

where $\mathbf{s}_{\alpha}, \mathbf{s}_{\beta}$ are the TCR $\alpha$ and $\beta$ chains, $\mathbf{s}_{\mathrm{pep}}$ is the presented peptide, and $\mathbf{s}_{\mathrm{MHC}}$ comprises MHC chains (heavy chain and $\beta_2$-microglobulin for Class I, or $\alpha/\beta$ chains for Class II).

Concatenated TCR-pMHC complexes range from 700 to 900 tokens (mean 780 Class I, 830 Class II), within ESM-2's 1024-token context. We apply no truncation to functional regions, right-pad shorter sequences, and exclude padding from the loss. No separators are inserted between chains.

ESM-2's positional embeddings span the concatenation continuously, and component-specific masking supplies the chain-boundary signal.

## 3.2. Two-Stage Continual Pre-training Curriculum

We adopt a continual pre-training strategy organized as a two-stage curriculum, progressing from large-scale pMHC and TCR sequence pre-training to paired TCR–pMHC interaction pre-training. Rather than uniform masking, we use functionally informed stage specific masking schedule that concentrates self-supervision on binding-relevant positions.

**Stage 1: Component-specific representations.** We train jointly on TCR repertoires from OTS (Raybould et al., 2024) and pMHC ligandomes from MHC Motif Atlas (Tadros et al., 2023).

The Stage 1 objective is:

$$\mathcal{L}_1(\theta) = \mathbb{E}_{x \sim \mathcal{D}_{\text{TCR}} \cup \mathcal{D}_{\text{pMHC}}} \mathbb{E}_{\mathcal{M} \sim \mathcal{P}_{\text{informed}}} \left[ \sum_{i \in \mathcal{M}} - \log p_\theta(x_i \mid \tilde{x}^{(\mathcal{M})}) \right] \quad (2)$$

where $\mathcal{P}_{\text{informed}}$ is a position-dependent masking distribution based on component specific functional annotations.

**Stage 2: Cross-chain interaction learning.** We continue training on paired TCR-pMHC sequences from VD-Jdb (Bagaev et al., 2020). The key insight is that paired data allows the model to learn how CDR sequences and peptide sequences constrain each other through the binding interface. We use joint masking of CDR and peptide positions:

$$\mathcal{L}_2(\theta) = \mathbb{E}_{\mathbf{z} \sim \mathcal{D}_{\text{paired}}} \mathbb{E}_{\mathcal{M} \sim \mathcal{P}_{\text{joint}}} \left[ \sum_{i \in \mathcal{M}} - \log p_\theta(z_i \mid \tilde{\mathbf{z}}^{(\mathcal{M})}) \right] \quad (3)$$

where $\mathcal{P}_{\text{joint}}$ is a joint masking distribution spanning all components. See Appendix B.1 for masking details and Appendix B.2 for optimization configuration.

This Stage 2 objective subsumes Stage 1 tasks. When peptide positions are masked, the model predicts them from MHC context. When CDR positions are masked, the model predicts them from framework context. The additional signal comes from cross-component conditioning, where CDR predictions leverage peptide context and vice versa. This compositional structure provides implicit experience replay, maintaining Stage 1 competencies while learning interaction-specific features.

## 3.3. Span Pseudo-Log-Likelihood Scoring

Standard pseudo-log-likelihood (PLL) masks single positions independently. For binding prediction, we instead use *span* PLL (sPLL), which masks entire functional regions simultaneously (Joshi et al., 2020). This better captures that

binding interfaces function as coherent units. We derive two uses of sPLL: *aggregate scores* for binding prediction and *per-position scores* for structural interpretation.

**Aggregate sPLL for binding prediction.** To predict binding compatibility, we mask all positions in a functional span and average their log-probabilities. Let $\mathcal{M}_{\text{pep}}$ denote the set of peptide position indices. For pMHC binding:

$$\text{sPLL}_{\text{pMHC}}(\mathbf{x}) := \frac{1}{|\mathcal{M}_{\text{pep}}|} \sum_{i \in \mathcal{M}_{\text{pep}}} \log p_\theta(x_i \mid \tilde{\mathbf{x}}^{(\mathcal{M}_{\text{pep}})}) \quad (4)$$

where $\tilde{\mathbf{x}}^{(\mathcal{M}_{\text{pep}})}$ denotes the pMHC sequence with peptide positions replaced by `[MASK]` and MHC positions unchanged. Following Goyal et al. (2021), higher sPLL indicates sequences more compatible with the learned distribution, though sPLL does not correspond to a normalized probability.

For TCR-pMHC recognition, we include TCR context:

$$\text{sPLL}_{\text{TpM}}(\mathbf{z}) := \frac{1}{|\mathcal{M}_{\text{pep}}|} \sum_{i \in \mathcal{M}_{\text{pep}}} \log p_\theta(z_i \mid \tilde{\mathbf{z}}^{(\mathcal{M}_{\text{pep}})}) \quad (5)$$

where peptide positions $\mathcal{M}_{\text{pep}}$ are indexed identically in both $\mathbf{x}$ and $\mathbf{z} = [\mathbf{x} \parallel \mathbf{y}]$.

**Per-position sPLL for structural interpretation.** To identify *which residues* drive predictions, we decompose sPLL into per-position contributions inspired by pointwise mutual information (PMI). For consistent distributions, $\text{PMI}(x; y) = \log p(x \mid y) - \log p(x)$ quantifies information gain from conditioning. Since masked language models do not define consistent joints (Goyal et al., 2021), we use a pseudo-likelihood analogue comparing predictions under different contexts.

**Intuition for pMHC binding.** The pMHC interaction score measures whether MHC context improves peptide prediction over background. High scores indicate positions where the MHC groove constrains amino acid identity, which are typically anchor positions P2 and PΩ (C-terminus). For HLA-A*02:01, these anchors are characteristically leucine at P2 and valine at PΩ.

**Definition 3.1** (pMHC Interaction Score). For peptide position $i \in \mathcal{M}_{\text{pep}}$:

$$\text{IS}_{\text{pMHC}}^{(i)}(\mathbf{x}) := \log p_\theta(x_i \mid \tilde{\mathbf{x}}^{(\mathcal{M}_{\text{pep}})}) - \log p_{\text{ref}}(x_i) \quad (6)$$

where $p_{\text{ref}}$ is the uniform distribution over the 20 standard amino acids

**Intuition for TCR recognition.** The TCR-pMHC interaction score measures additional predictive information from TCR presence beyond MHC alone. High scores indicate

positions where TCR directly contacts the peptide, which are typically solvent-exposed central positions P4–P7 that engage CDR3 loops.

**Definition 3.2** (TCR-pMHC Interaction Score). For peptide position $i \in \mathcal{M}_{\text{pep}}$:

$$\text{IS}_{\text{TpM}}^{(i)}(\mathbf{z}) := \log p_\theta(z_i \mid \tilde{\mathbf{z}}^{(\mathcal{M}_{\text{pep}})}) - \log p_\theta(x_i \mid \tilde{\mathbf{x}}^{(\mathcal{M}_{\text{pep}})}) \tag{7}$$

Positive scores indicate positions where TCR context improves peptide prediction.

### 3.4. Inference: Iterative Entropy-Guided Refinement

Given a target pMHC and a TCR scaffold (fixed V/J genes and framework regions), we aim to design the Complementarity-Determining Regions (CDR), specifically the CDR3$\alpha$ and CDR3$\beta$ sequences which jointly confer specific recognition. This is a constrained generation problem where the designed regions must be compatible with both scaffold context and target antigen.

Standard approaches have limitations. Autoregressive generation imposes arbitrary left-to-right ordering that ignores bidirectional constraints from framework regions. Single-pass parallel decoding ignores dependencies between designed positions. We propose **Iterative Entropy-Guided Refinement (IEGR)**, adapting confidence-based non-autoregressive decoding (Chang et al., 2022; Ghazvininejad et al., 2019) to constrained protein design. The key insight is that resolving high-confidence positions first provides reliable context for subsequent predictions. We contrast IEGR with two simpler non-autoregressive baselines on the same masked-position setup. **Greedy** decoding selects the argmax token at every position in a single forward pass, with no sampling and no use of inter-position dependencies. **OneShot** decoding samples all positions simultaneously from the predicted distributions in a single pass, capturing marginal predictions but not the dependencies between designed residues. Both lack the iterative refinement and confidence-based ordering that IEGR provides.

**Biological motivation for entropy ordering.** Both CDR3$\alpha$ and CDR3$\beta$ loops contribute to the TCR-pMHC interface, with CDR3$\beta$ typically making dominant peptide contacts while CDR3$\alpha$ modulates specificity. Designing both loops jointly ensures interface compatibility. CDR3 loops share conserved boundary residues, including N-terminal cysteines and C-terminal Phe-Gly or Trp-Gly motifs, that anchor loop geometry. Central positions are hypervariable and determine antigen specificity.

Entropy-guided ordering naturally respects this hierarchy. Anchor positions have low predictive entropy and are resolved first, establishing a stable scaffold. High-entropy central positions are resolved last, conditioned on estab-lished anchors. By computing entropy across both loops simultaneously, the algorithm interleaves decoding between chains, selecting whichever position has lowest uncertainty at each step. This allows cross-chain dependencies to inform the design of both loops.

**Algorithm.** Let $\mathcal{U}_0$ denote design positions (CDR3 residues). For masked positions $\mathcal{M}_t$ at step $t$, we quantify uncertainty via predictive entropy:

$$\text{H}_\theta^t(s_i) = -\sum_{v \in \mathcal{V}} p_\theta(s_i = v \mid \tilde{\mathbf{s}}^{(\mathcal{M}_t)}) \log p_\theta(s_i = v \mid \tilde{\mathbf{s}}^{(\mathcal{M}_t)}) \tag{8}$$

**Phase 1: Entropy-Guided Construction.** Starting from all design positions masked, we iteratively select the position with lowest entropy (highest confidence), sample its value from the predicted distribution with temperature $\tau$, and remove it from the mask set. This continues until all positions are filled.

**Phase 2: Block Gibbs Refinement.** After initial construction, we perform $K$ rounds of local refinement. Each round selects a random block of $b$ positions, masks them, and resamples. This enables escape from suboptimal configurations.

Since masked language model conditionals need not arise from a consistent joint distribution (Goyal et al., 2021), IEGR lacks classical MCMC guarantees. However, the entropy-guided ordering can be interpreted as a greedy approximation to maximizing mutual information between resolved and unresolved positions at each step. We view IEGR as a practical search heuristic whose effectiveness we evaluate empirically. Full pseudocode appears in Algorithm 1; implementation details and hyperparameters are provided in Appendix C.

## 4. Experimental Setup

**Training Data.** Stage 1 trains on TCR repertoires ($1.5 \times 10^6$ paired TCR $\alpha\beta$ sequences from OTS (Raybould et al., 2024)) and pMHC ligandomes ($1.0 \times 10^6$ peptide-MHC pairs from two large monoallelic cell line datasets, and the MHC Motif Atlas (Tadros et al., 2023; Sarkizova et al., 2020; Abelin et al., 2017)) combined with synthetic peptides consisting of high-confidence predictions from MixMHCpred-2.2 (Gfeller et al., 2023). Stage 2 trains on paired TCR-pMHC interactions ($3.0 \times 10^4$ from VDJdb (Bagaev et al., 2020)). Optimization hyperparameters and computational cost are detailed in Appendix B.

**Evaluation Splits.** We use strict held-out splits to evaluate generalization. For pMHC binding, we held out 10 Class I alleles and 6 Class II alleles completely from Stage 1. See Table 7 for a complete list. For TCR-pMHC recognition, we

**Algorithm 1** Iterative Entropy-Guided Refinement (IEGR)

---

**Require:** Template $\mathbf{z} = [\mathbf{x} \parallel \mathbf{y}]$, design positions $\mathcal{U}_0$, refinement rounds $K$, block size $b$, temperature $\tau$, model $p_\theta$
**Ensure:** Designed sequence set $S$
1: $\mathbf{s} \leftarrow \mathbf{z}$
2: $\mathbf{s}_i \leftarrow$ [MASK] for all $i \in \mathcal{U}_0$
3: $\mathcal{U} \leftarrow \mathcal{U}_0$ {Remaining masked positions}
  *// Phase 1: Entropy-Guided Construction*
4: **while** $\mathcal{U} \neq \emptyset$ **do**
5:     **for** $i \in \mathcal{U}$ **do**
6:         $\pi_i(v) \leftarrow p_\theta(s_i = v \mid \tilde{s}^{(\mathcal{U})})$ for all $v \in \mathcal{V}$
7:         $\mathrm{H}_\theta(s_i) \leftarrow -\sum_v \pi_i(v) \log \pi_i(v)$
8:     **end for**
9:     $i^* \leftarrow \arg\min_{i \in \mathcal{U}} \mathrm{H}_\theta(s_i)$ {Lowest entropy}
10:     $\mathbf{s}_{i^*} \sim \mathrm{Softmax}(\mathrm{logits}_{i^*}/\tau)$
11:     $\mathcal{U} \leftarrow \mathcal{U} \setminus \{i^*\}$
12: **end while**
  *// Phase 2: Block Gibbs Refinement*
13: **for** $k = 1$ **to** $K$ **do**
14:     $\mathcal{B} \leftarrow$ random block of size $b$ positions from $\mathcal{U}_0$
15:     $\mathbf{s}_i \leftarrow$ [MASK] for all $i \in \mathcal{B}$
16:     **for** $i \in \mathcal{B}$ **do**
17:         $\mathbf{s}_i \sim p_\theta(\cdot \mid \tilde{s}^{(\mathcal{B})})$ at temperature $\tau$
18:     **end for**
19:     **if** $k \mod |\mathcal{U}_0| = 0$ **then**
20:         $S \leftarrow S \cup \{s\}$
21:     **end if**
22: **end for**
23: **return** $S$

---

held out epitopes YLQPRTFLL (SARS-CoV-2 Spike) and GLCTLVAML (EBV BMLF1) and all TCRs recognizing these epitopes from Stage 2. Full details in Appendix D.

**Models and Baselines.** DecoderTCR uses ESM-2 backbones at 650M and 3B parameters. For pMHC binding, we compare against MHCflurry 2.0 (O'Donnell et al., 2020), MixMHCpred-3.0 (Tadros et al., 2025), NetMHCpan-4.1 (Reynisson et al., 2020). Zero-shot ESM scoring is performed similarly as Eq. 4. For TCR-pMHC recognition, we compare against NetTCR-2.2 (Jensen & Nielsen, 2023). For generation evaluation, we use AlphaFold3 (Abramson et al., 2024) interface pTM (ipTM) as an in silico oracle.

**In silico design evaluation.** We evaluate generated CDR3$\alpha/\beta$ designs along five metrics: (1) AlphaFold3 ipTM (Abramson et al., 2024) as the primary structural oracle along with (2) Boltz-2 ipTM (Passaro et al., 2025), (3) OLGA $\log P_{\mathrm{gen}}$ (Sethna et al., 2019) for V(D)J recombination plausibility, (4) salt-bridge enrichment for antigen-specific contacts, and (5) $k$-mer spectrum shift for sequence naturalness. We further compare the three decod-

ing strategies (IEGR, Greedy, OneShot) against a structure-conditioned baseline pipeline. For this baseline, we use RFdiffusion (Watson et al., 2023) for backbone generation followed by ProteinMPNN (Dauparas et al., 2022) for sequence design, conditioned on PDB 7N6E as the structure template and producing 100 backbones with 10 Protein-MPNN samples each, yielding 1,000 designs. Validated binding and non-binding YLQ TCRs from Messemaker et al. (2025) serve as empirical reference distributions.

## 5. Results

We evaluate DecoderTCR across representation quality, knowledge retention, binding prediction, structural interpretability, and experimentally validated *de novo* design.

### 5.1. pMHC Binding Prediction

Following Stage 1 training on marginal pMHC ligandomes and TCR repertoires with informed masking, we assess model quality along two dimensions: (1) zero-shot generalization to unseen HLA alleles and (2) representation transferability for supervised pMHC binding prediction, both prior to Stage 2 training.

**Zero-shot generalization.** We rank peptides by aggregate sPLL (Eq. 4). Table 1 reports performance on 16 HLA alleles excluded from training. ESM-2, despite training on billions of protein sequences, performs near random (AUROC 0.41-0.56), confirming that generic protein representations do not encode MHC-specific binding preferences. Stage 1 DecoderTCR achieves AUROC 0.93-0.96 on Class I and 0.90-0.91 on Class II, demonstrating that domain-specific pretraining yields strong zero-shot transfer. The modest Class I versus Class II gap reflects biological differences in binding groove conservation (Jones et al., 2006). Per-allele ROC and PR curves are shown in Appendix Figure 5.

*Table 1.* **Zero-shot pMHC binding.** Stage 1 trains jointly on pMHC and TCR data. Test alleles completely excluded from training. Mean $\pm$ SD across 10 Class I and 6 Class II alleles.

| Model | Class I | | Class II | |
|---|---|---|---|---|
| | AUROC | AUPRC | AUROC | AUPRC |
| *Generic Baselines* | | | | |
|   ESM-2 (650M) | .41±.09 | .43±.05 | .51±.22 | .54±.17 |
|   ESM-2 (3B) | .45±.11 | .46±.06 | .56±.18 | .56±.15 |
| *DecoderTCR (Stage 1)* | | | | |
|   650M | **.96**±.04 | **.96**±.05 | .90±.02 | .89±.02 |
|   3B | .93±.08 | .92±.10 | **.91**±.02 | **.90**±.03 |

**Transfer to supervised prediction.** To verify that representations encode transferable features, we train an MLP

classifier on frozen embeddings using IEDB $pIC_{50}$ binding affinity data and evaluate on ESCAPE-seq (Shi et al., 2025), an independent prospective benchmark. DecoderTCR-3B achieves AUROC 0.77, matching specialized predictors (MixMHCpred, NetMHCpan, MHCflurry) that were purpose-built with allele-specific architectures. ESM-2 achieves only 0.62-0.63 regardless of scale, confirming that model size alone does not substitute for domain-relevant pretraining (Table 2).

*Table 2.* **Supervised Oncogene mutant pMHC binding prediction.** MLP classifier on frozen Stage 1 embeddings, tested on prospective data from Shi et al. (2025).

| Model | AUROC | AUPRC |
|---|---|---|
| *Specialized Predictors* | | |
| MixMHCpred-3.0 | .77±.07 | .44±.15 |
| NetMHCpan-4.1 | .77±.08 | .44±.15 |
| MHCflurry 2.0 | .76±.08 | .45±.14 |
| *ESM Embeddings* | | |
| ESM-2 (650M) | .62±.06 | .19±.12 |
| ESM-2 (3B) | .63±.07 | .19±.12 |
| *DecoderTCR Embeddings (Stage 1)* | | |
| 650M | .75±.07 | .35±.15 |
| 3B | **.77**±.07 | **.35**±.16 |

## 5.2. Knowledge Retention During Continual Learning

A central question for our two-stage continual pre-training curriculum is whether Stage 2 training on paired data overwrites Stage 1 representations. We monitor two sentinel metrics throughout Stage 2 training progression: zero-shot pMHC binding AUROC on held-out alleles and TCR inverse perplexity on held-out TCR-pMHC sequences.

Figure 2 demonstrates both metrics remain stable or improve throughout 3500 steps of Stage 2 training. Zero-shot pMHC AUROC remains constant at 0.925-0.93 with no discernible degradation. TCR inverse perplexity increases from 0.72 to 0.79 (+10%), suggesting that exposure to binding-relevant CDR sequences during Stage 2 refines rather than overwrites TCR representations. We posit this stability arises from *implicit experience replay*: each paired TCR-pMHC example contains valid pMHC and TCR subsequences, so the joint masking objective (Eq. 3) subsumes Stage 1 tasks while adding cross-component signal.

## 5.3. Epitope-Specific TCR Recognition

We next evaluate whether DecoderTCR can distinguish binding from non-binding TCRs for a fixed pMHC target, the setting most relevant to annotating TCRs from patient repertoires for diagnostic purposes and to selecting TCR candidates for therapeutic design.

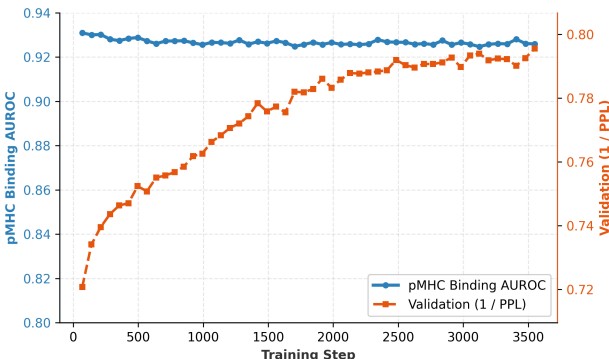

*Figure 2.* **Knowledge retention during Stage 2.** Zero-shot pMHC AUROC (blue) measures binding prediction on alleles excluded from all training. TCR inverse perplexity (orange) measures language modeling quality on held-out sequences.

**Evaluation protocol.** We use YLQPRTFLL (YLQ; SARS-CoV-2 Spike) and GLCTLVAML (GLC; EBV BMLF1), both HLA-A*02:01-restricted, with experimentally validated binders and non-binders from Messemaker et al. (2025). We prioritize these epitopes for ground-truth reliability over breadth, given that most public TCR-pMHC labels remain noisy. TCR-pMHC pairs are scored by summed sPLL over masked peptides under the joint context, where higher scores indicate higher predicted binding probability. Both epitopes are completely excluded from all DecoderTCR training, enabling true zero-shot evaluation.

**Zero-shot DecoderTCR approaches supervised baselines.** Table 3 reports AUROC against five supervised methods trained explicitly on these epitopes. Despite never seeing YLQ or GLC during training, DecoderTCR-3B achieves AUROC 0.76 on YLQ and 0.64 on GLC, approaching the strongest supervised baseline NetTCR-2.2 (0.88 and 0.69) and exceeding four other supervised methods. Stage ablations and AUPRC values appear in Appendix E.2, Table 9. ESM-2 performs near chance, confirming that domain-specific pretraining rather than scale drives recognition.

**Ablation summary.** Both pre-training stages are necessary: Stage 1 alone fails (0.18 AUROC on YLQ) because pMHC and TCR representations are not aligned without paired supervision, and Stage 2 alone underperforms (0.56 AUROC) without the regularization from unpaired data. The full pipeline reaches 0.76 AUROC. Stage ablation and masking-strategy ablation results appear in Appendix E. For inference, entropy-guided ordering outperforms single-pass decoding (Section 5.5).

## 5.4. Structural Interpretability via Interaction Scores

While aggregate sPLL enables binding prediction, it does not reveal *which residues* drive the prediction. We apply

*Table 3.* **Epitope-specific TCR recognition.** AUROC for binding vs. non-binding TCR classification. Both epitopes are excluded from DecoderTCR training.

| Model | YLQ | GLC |
|---|---|---|
| *Supervised* | | |
| NetTCR-2.2 | .88 | .69 |
| PanPep | .65 | .49 |
| ERGO-II | .52 | .64 |
| pMTnet | .58 | .50 |
| DLpTCR | .47 | .54 |
| *Zero-shot* | | |
| ESM-2 (3B) | .36 | .48 |
| DecoderTCR (3B) | **.76** | **.64** |

per-position interaction scores (Eqs. 6–7) to the held-out validated YLQPRTFLL positive binders and overlay the summarized interaction scores over a TCR-pMHC complex with solved structure (PDB 7RTR), using no structural supervision during training. By treating the peptide as the shared interface, we decompose binding signal into MHC-driven anchoring versus TCR-driven recognition.

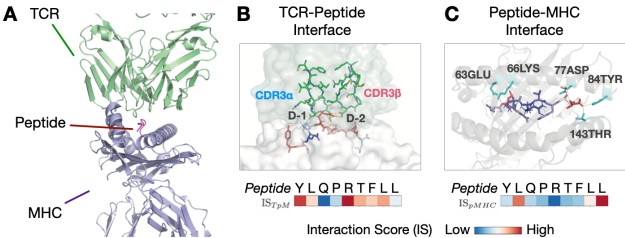

*Figure 3.* **Interaction scores recover structural contacts without supervision.** (A) Overview of TCR-pMHC ternary complex (PDB 7RTR) showing TCR (green) engaging peptide (pink) presented by MHC (purple). (B) TCR-peptide interface with two salt bridges between TCR$\alpha$ Asp109/TCR$\beta$ Asp110 and peptide Arg5. $IS_{TpM}$ highlights TCR contact positions P4-P7. (C) Peptide-MHC interface with anchor residues P2 (Leu) and P9 (Leu) buried in MHC pockets. $IS_{pMHC}$ highlights anchors P2 and P9.

**TCR-Peptide interface.** $IS_{TpM}$ (Eq. 7) quantifies how TCR context improves peptide prediction beyond MHC alone. Positions P4-P7 show elevated scores, corresponding to the peptide bulge contacting CDR3 loops (Figure 3B). Anchor positions P2 and P9 show minimal TCR-dependent signal, as expected for residues facing the MHC groove rather than TCR.

**Peptide-MHC interface.** $IS_{pMHC}$ (Eq. 6) quantifies how MHC context improves prediction at each peptide position. Positions P2 and P9 show elevated scores, corresponding to canonical anchor residues buried in MHC binding pockets (Figure 3C). These anchors (typically leucine at P2 and

valine at P9 for HLA-A*02:01) drive stable peptide presentation (Madden et al., 1993). Central positions P4-P7 show lower MHC-dependent signal.

The complementary pattern (MHC anchors at termini, TCR contacts at center) emerges purely from sequence-level training, suggesting representations capture biologically relevant interface features.

### 5.5. Generative Design and Experimental Validation

Moving from binding prediction to sequence design, we evaluate the model's ability to zero-shot generate novel binders for a strictly held-out target. Given YLQPRTFLL/HLA-A*02:01 and a TCR scaffold (fixed V/J genes and framework regions taken from validated positive binding TCRs in (Messemaker et al., 2025)), the task is to design CDR3$\alpha$ and CDR3$\beta$ sequences that confer target recognition in a zero-shot setting. We evaluate all three decoding strategies described in Section 3.4 (IEGR, Greedy, OneShot).

**In silico evaluation.** Figure 4 compares strategies using AlphaFold3 ipTM as a structural proxy. Single-pass methods (Greedy, OneShot) produce ipTM distributions comparable to validated non-binders. IEGR significantly outperforms both (Mann-Whitney $p < 0.001$) by resolving low-entropy anchors before high-entropy positions, enabling escape from suboptimal configurations. Beyond AF3 ipTM, IEGR leads on the four additional metrics defined in Section 4 (Table 11 in Appendix H), including 34.2% salt-bridge enrichment compared with 17.2% for Greedy and 21.9% for OneShot. The RFdiffusion + ProteinMPNN baseline, despite access to a solved crystal structure, produces sequences nearly 10 log-units outside the natural repertoire on OLGA, underscoring that structural plausibility without immune-specific sequence priors is insufficient for TCR design.

**Lab-in-the-loop experimental validation.** We tested designs *in vitro* across two iterative rounds, using Round 1 to characterize the prediction-generation gap and Round 2 to narrow it with model-guided filtering. In Round 1, we synthesized 35 designs (Appendix F, Table 10), expressed them in TCR-knockout Jurkat cells, and measured dextramer binding by flow cytometry (Appendix G). One IEGR design bound above background (1/35, ~3%). Greedy and OneShot designs yielded none. This confirmed that sequence-first design can produce a *de novo* binder for a held-out target, but also exposed a clear gap: high *in silico* confidence did not reliably predict binding feasibility in this context.

For Round 2, we trained an $L_2$-regularized logistic regression classifier on frozen DecoderTCR-3B embeddings, us-

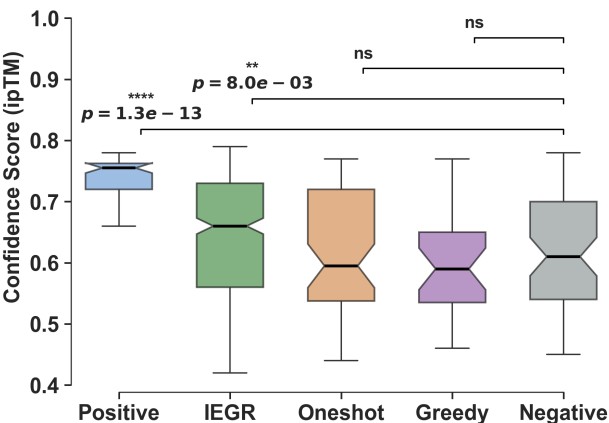

*Figure 4.* ***In silico*** **assessment of CDR design quality.** Distributions of AlphaFold3 ipTM scores for designs targeting YLQPRTFLL/HLA-A*02:01. Positive and Negative controls are taken from (Messemaker et al., 2025) validated YLQ binding TCRs. Brackets denote two-sided Wilcoxon rank-sum tests against validated negative controls. IEGR significantly outperforms non-binders ($p < 0.001$).

ing validated YLQ binders from Messemaker et al. (2025) plus the Round 1 hit as positives ($n$=66), and validated non-binders plus Round 1 failed designs as negatives ($n$=44). The Round 1 failures act as hard negatives: they scored well on AF3 ipTM yet did not bind, forcing the classifier to learn distinctions the structural oracle missed. We synthesized 25 new IEGR designs prioritized by the classifier and tested them *in vitro*: 3/25 (12%) bound above background, a $\sim 4\times$ improvement over Round 1. The prediction-generation gap is progressively bridgeable through iterative experimental feedback, though closing it without wet-lab signal remains an open challenge.

## 6. Discussion and Conclusion

We introduced a compositional pretraining framework demonstrating that protein language models *can* effectively model multi-component immune interfaces. Our two-stage curriculum provides a template for foundation models in domains with heterogeneous data availability.

**The prediction-generation gap.** Our results reveal that strong zero-shot *prediction* does not trivially translate to reliable *generation*. The model discriminates binders from non-binders substantially better than chance (0.76 AUROC) but is not yet capable of consistently generating strong binders. This asymmetry suggests that ranking requires capturing *necessary* conditions (sequence grammar, anchor constraints), while generation requires *sufficient* conditions demanding more paired data or thermodynamic calibration. However, a simple classifier trained on DecoderTCR embed-

dings increased the experimental hit rate from 3% to 12% in a second screening round, demonstrating that the representations encode actionable biochemical signal not captured by structural confidence scores alone. This establishes a lab-in-the-loop paradigm in which each experimental round generates training signal that refines the next round of generative selection.

**Broader implications.** DecoderTCR's contributions extend beyond TCR-pMHC in two ways. First, compositional pre-training enables sample-efficient learning by projecting interaction learning onto a lower-dimensional manifold where limited paired supervision suffices. Second, confidence-based decoding respects biological hierarchy, naturally resolving conserved anchors before hypervariable positions. These principles may apply broadly to multi-component interfaces where paired data is scarce but marginal data is abundant. Together with experimental approaches that *read* TCR specificity, such as through deep peptide recognition profiling (Wang et al., 2026), DecoderTCR addresses the complementary problem of *writing* specificity by generating TCRs with desired antigen recognition from sequence alone, advancing a read/write framework for engineered T-cell immunity.

**Limitations.** Zero-shot epitope-specific design remains challenging. While all designed sequences are novel (edit distance $\geq 3$), we used existing binders as scaffolds; scaffold-free design is unsolved. Wet-lab validation was performed for one pMHC target, although *in silico* generation metrics extend to a second held-out epitope (GLCTLVAML; Appendix H). Recognition performance varies across antigens, reflecting fundamental data sparsity (Lu et al., 2026). Recent estimates suggest $10^6$–$10^8$ epitopes may be required for fully generalizable prediction (Delaunay et al., 2025); our approach may reduce but does not eliminate this data requirement. Finally, because DecoderTCR learns the joint distribution of the immune synapse, it can in principle generate peptides conditioned on a fixed TCR scaffold, a direction we leave for future work.

**Conclusion.** DecoderTCR demonstrates that compositional pre-training enables strong zero-shot *prediction* for TCR-pMHC interfaces, substantially outperforming generic protein language models. For *generation*, IEGR produces sequences passing structural and biological plausibility filters, and a lab-in-the-loop iteration improves the experimental hit rate from 3% to 12%, demonstrating that the prediction-generation gap for the immune synapse can be systematically bridged. To our knowledge, this work provides the first explicit characterization of this gap and offers a concrete, validated strategy to narrow it, establishing a robust benchmark for generative immunology.

## Impact Statement

This work contributes to computational immunology with the goal of accelerating new therapies and diagnostics for cancer, autoimmunity, and infectious diseases. The ability to generate high-affinity TCRs *in silico* could reduce drug discovery time and cost, making personalized immunotherapies more accessible.

We acknowledge dual-use potential: generative capabilities for targeting cancer neoantigens could theoretically be misused to design receptors targeting healthy tissue or benign antigens. Additionally, training data is historically skewed toward MHC alleles common in European populations, risking inequitable distribution of therapeutic utility.

To mitigate these risks, we advocate for: (1) rigorous *in vitro* off-target screening against healthy tissue antigens before clinical application; (2) community prioritization of diverse immunogenomic datasets for equitable allele coverage; and (3) responsible model release under licenses restricting use to therapeutic research. We believe that with these safeguards, the therapeutic benefits of generative immunology outweigh the risks.

## Code and Data Availability

Code is available at `https://github.com/czbiohub-chi/DecoderTCR`, and model checkpoints and supporting files are hosted at `https://huggingface.co/biohub/DecoderTCR`.

## Acknowledgements

We thank Kavita Kulkarni, Amanda Surya, Shaowen Zhu, Kyle Hippe, and Tim Rand for helpful discussions. A.A.K. was supported in part by NIH DP2AI177884, a Chan Zuckerberg Investigator Award, and a Breakthrough T1D/LRA/NMSS Common Mechanisms of Autoimmunity Insight Award (1-SRA-2025-1755-A-N).

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

## A. Biological Background

For the general machine learning audience, we define key biological terms and concepts.

**The TCR-pMHC Interface.** **T-cell receptors (TCRs)** recognize short protein fragments (**peptides**) presented on cell surfaces by the **Major Histocompatibility Complex (MHC, or HLA in humans)**. This interaction drives adaptive immunity and forms the basis of targeted T-cell therapies. TCRs consist of $\alpha$ and $\beta$ chains, each containing three **Complementarity-Determining Regions (CDRs)**. While CDR1 and CDR2 primarily contact the MHC scaffold, the hypervariable **CDR3** loop directly engages the peptide. CDR3 diversity is driven by **V(D)J recombination**, which joins Variable (V), Diversity (D, $\beta$-chain only), and Joining (J) gene segments with imprecise junctions. Because CDR3 dictates antigen specificity, our generative task holds the V/J framework constant and exclusively optimizes the CDR3$\alpha$ and CDR3$\beta$ loops.

**MHC Restriction and Anchoring.** MHC molecules are divided into **Class I** (presenting 8–14 residue peptides to CD8+ T-cells) and **Class II** (12–21 residues to CD4+ T-cells). Peptides bind the MHC groove via side-chain interactions at conserved **anchor** positions. For example, in HLA-A*02:01, positions P2 and P$\Omega$ (the C-terminus) strongly anchor the peptide, leaving central residues (P4–P7) solvent-exposed for TCR contact. Figure 3 demonstrates DecoderTCR's ability to recover this structural dichotomy purely from sequence. Furthermore, recognition is strictly conditioned on the MHC allele (**MHC restriction**), meaning a TCR specific to a peptide on one allele will generally fail to recognize it on another.

**The Recognition Problem.** TCR-pMHC recognition is a joint condition over four components: TCR$\alpha$, TCR$\beta$, peptide, and MHC. The mapping is many-to-many: a single TCR can recognize multiple peptides (**degeneracy**), and a single peptide can be bound by diverse TCRs (**polyspecificity**). This complex dependency structure necessitates the unified multi-chain representation introduced in Section 3.1.

**The Data Landscape.** Public data for these components exhibits severe asymmetry. While unpaired TCR $\alpha\beta$ repertoires ($\sim 10^7$ sequences; Raybould et al. 2024) and eluted pMHC ligandomes ($\sim 10^6$ pairs; Tadros et al. 2023; Sarkizova et al. 2020; Abelin et al. 2017) are relatively abundant, validated TCR-pMHC interactions are scarce ($< 10^5$ pairs; Bagaev et al. 2020) and solved ternary structures are exceptionally rare ($< 500$ complexes). This four-order-of-magnitude hierarchy motivates our compositional pre-training strategy: abundant marginal data constrains individual component representations, while scarce paired data refines the cross-chain interface.

## B. Training Details

This appendix provides implementation details for the training procedures described in Section 3.2.

### B.1. Stage-Specific Masking Distributions

We use a masked residue prediction objective with a component- and stage-dependent masking distribution. For a sequence $x$, we sample a mask set $M \subseteq \{1, \ldots, |x|\}$ by independently masking position $i$ with probability $p(i)$ determined by its annotation (e.g., peptide, HLA pocket, CDR, or other). To take advantage of the structure of the interaction modality of the system of interest, we define:

$$\mathcal{L}_s(\theta) = \mathbb{E}_{x \sim \mathcal{D}_s} \, \mathbb{E}_{M \sim \mathcal{M}_s} \left[ \sum_{i \in M} \left( -\log p_\theta(x_i \mid x_{\setminus M}) \right) \right] \tag{9}$$

**Stage 1 (Component specific).** Stage 1 uses two datasets: a peptide–MHC (pMHC) dataset $\mathcal{D}_{\mathrm{pMHC}}$ and an unpaired TCR dataset $\mathcal{D}_{\mathrm{TCR}}$. For samples $x \sim \mathcal{D}_{\mathrm{pMHC}}$, we define the per-position masking probabilities as

$$p_{\mathrm{pMHC}}(i) = \begin{cases} 0.50, & i \in \mathrm{Peptide}(x), \\ 0.20, & i \in \mathrm{HLA \ Pocket}(x), \\ 0.10, & \text{otherwise,} \end{cases} \tag{10}$$

where $\mathrm{Pocket}(x)$ denotes the HLA binding pocket residues(exon2 & exon3). For samples $x \sim \mathcal{D}_{\mathrm{TCR}}$, we use

$$p_{\mathrm{TCR}}(i) = \begin{cases} 0.50, & i \in \mathrm{CDR}(x), \\ 0.10, & \text{otherwise.} \end{cases} \tag{11}$$

We sample $M$ by masking each position $i$ independently with the corresponding probability $p_{\mathrm{pMHC}}(i)$ or $p_{\mathrm{TCR}}(i)$, denoted $\mathcal{P}_{\mathrm{informed}}$.

**Stage 2 (Cross-component interaction).** Stage 2 trains on a paired TCR–pMHC interaction dataset $\mathcal{D}_{\mathrm{paired}}$. For interaction samples $x \sim \mathcal{D}_{\mathrm{paired}}$, we define

$$p_{\mathrm{int}}(i) = \begin{cases} 0.20, & i \in \mathrm{HLA\ Pocket}(x), \\ 0.50, & i \in \mathrm{Peptide}(x)\ \cup\ \mathrm{CDR}(x), \\ 0.10, & \mathrm{otherwise}. \end{cases} \tag{12}$$

As in Stage 1, we sample $M$ by independently masking each position $i$ with probability $p_{\mathrm{int}}(i)$.

### B.2. Optimization Configuration

We train with AdamW optimizer (learning rate $5 \times 10^{-5}$, $\beta_1 = 0.9$, $\beta_2 = 0.999$, weight decay 0.01). Batch size 4 per device across 96 H100 GPUs with BF16 mixed precision. Stage 1 processes $5.03 \times 10^{11}$ tokens. Stage 2 runs for 20 epochs. IEGR hyperparameters: $K = 5 \times |\mathcal{U}_0|$ refinement rounds, block size $b = 3$, temperature $\tau = 0.1$. Hyperparameters selected on validation set (10% of paired data).

### B.3. Computational Cost

Model training processes around $5.03 \times 10^{11}$ tokens required approximately 4,500 GPU hours on NVIDIA H100 Tensor Core GPUs, corresponding to roughly $1.6 \times 10^{22}$ FLOPs ($\approx 1.6 \times 10^{10}$ TFLOPs, or 16 exaFLOPs, BF16-equivalent).

## C. IEGR Algorithm Details

This section provides implementation details for Algorithm 1 in Section 3.4.

### C.1. Hyperparameters

*Table 4.* IEGR hyperparameters.

| Parameter | Value | Rationale |
|---|---|---|
| Temperature $\tau$ | 0.1 | Low temperature reduces sampling noise while maintaining diversity |
| Block size $b$ | [1, 3, 5] | Balances local refinement with escape from local optima |
| Refinement rounds $K$ | $5 \times |\mathcal{U}_0|$ | Ensures each position revisited multiple times |

### C.2. Implementation Details

**Entropy computation.** For masked positions $\mathcal{M}$, predictive entropy quantifies uncertainty:

$$\mathrm{H}_\theta(s_i) = -\sum_{v \in \mathcal{V}} p_\theta(s_i = v \mid \tilde{\mathbf{s}}^{(\mathcal{M})}) \log p_\theta(s_i = v \mid \tilde{\mathbf{s}}^{(\mathcal{M})}) \tag{13}$$

where $\mathcal{V}$ is the 20-amino-acid vocabulary. We use natural logarithm, giving entropy in nats with maximum $\log(20) \approx 3.0$.

**Temperature-scaled sampling.** Given logits $\ell_i \in \mathbb{R}^{|\mathcal{V}|}$ for position $i$, we sample:

$$s_i \sim \mathrm{Categorical}\left(\mathrm{Softmax}(\ell_i / \tau)\right) \tag{14}$$

Lower $\tau$ concentrates probability mass on high-likelihood residues.

**Block selection.** In Phase 2 refinement, blocks are sampled uniformly at random from design positions $\mathcal{U}_0$. Positions within a block are resampled with temperature $\tau$.

**Motivation for Entropy-Guided Decoding Scheduling.** Standard MCMC sampling on protein landscapes often suffers from slow mixing and kinetic trapping. Our IEGR schedule is a heuristic approximation inspired by confidence-based non-autoregressive decoding (Ghazvininejad et al., 2019). While we do not provide theoretical guarantees on mode coverage or ergodicity, the schedule is motivated by the biophysical hierarchy of protein folding: high-confidence residues (anchors) effectively constrain the conformational search space for lower-confidence loops. Empirically, we find this "confidence-first" filling reduces the likelihood of hallucinating invalid high-energy states early in the generation process compared to random-order Gibbs sampling.

### C.3. Hyperparameter Sensitivity

We evaluated IEGR sensitivity to refinement rounds ($K$), block size ($b$), and sampling temperature ($T$) on YLQPRTFLL/HLA-A*02:01 designs. IEGR is robust across the tested range (Table 5); reducing refinement from $K=5$ to $K=1$ produces the largest AF3 ipTM decrease, while block size and temperature changes produce only modest shifts without catastrophic failure.

*Table 5.* IEGR hyperparameter sensitivity on YLQPRTFLL/HLA-A*02:01 designs. Default setting is $K=5$, $b=3$, $T=0.1$.

| Setting | AF3 ipTM | Boltz-2 ipTM | OLGA | Salt-bridge | $k$-mer shift |
|---------|----------|--------------|------|-------------|---------------|
| Default | .655 | .845 | -8.25 | 29.6% | .594 |
| $K=1$ | .634 | .845 | -8.16 | 29.6% | .603 |
| $K=10$ | .653 | .843 | -8.36 | 32.3% | .587 |
| $b=1$ | .639 | .844 | -8.42 | 27.3% | .602 |
| $b=5$ | .655 | .843 | -8.27 | 28.1% | .592 |
| $T=0.3$ | .654 | .845 | -8.23 | 29.0% | .592 |
| $T=0.5$ | .646 | .845 | -8.53 | 27.6% | .592 |

### C.4. Variable-Length CDR3 Generation

Our main experiments fix CDR3$\alpha$ and CDR3$\beta$ lengths to match the scaffold, isolating sequence design from loop-length design. We also tested nearby CDR3 length variants *in silico*. AF3 ipTM decreased smoothly as designed loops moved away from the scaffold length, consistent with geometric constraints on CDR3 loop closure and interface positioning. We therefore focus the present work on fixed-length CDR3 redesign and leave joint length-and-sequence generation for future work.

## D. Data Preparation

This appendix details data collection and processing for the experimental setup described in Section 4.

### D.1. HLA Data Processing

HLA protein sequences were collected from the IPD-IMGT/HLA alignment database (Barker et al., 2023). Sequences were collected for 4 Class I genes (A, B, C, and E) and 6 Class II genes (DRA, DRB1, DPA1, DPB1, DQA1, DQB1). Eight-digit HLA allele names were simplified to four-digit allele resolution, then deduplicated based on unique protein sequences, keeping the longest sequence variant. A total of 17,065 HLA Class I alleles and 7,718 HLA Class II alleles were used for further processing.

### D.2. pMHC Data

#### D.2.1. EXPERIMENTALLY VALIDATED

We collected experimentally validated pMHC pairs from Sarkizova et al. (2020); Tadros et al. (2023); Abelin et al. (2017); Pyke et al. (2021) for Class I HLAs and from The MHC Motif Atlas (Tadros et al., 2023) for Class II HLAs. Class I pairs were further filtered using MixMHCpred (binding score $> -2$), and all samples were deduplicated based on peptide-allele pairs. We retained 451,828 unique pairs across 123 Class I HLA alleles and 548,858 unique pairs across 81 HLA Class II alleles.

D.2.2. SYNTHETIC

For synthetic pMHC binders, human proteome sequences from the NCBI RefSeq database using the GRCh38.p14 assembly (Accession: GCF-000001405.40) were used to generate new peptides. The peptides of varying length (8–14 for Class I; 12–21 for Class II) were generated using a sliding window and scored with MixMHCpred across the 126 Class I alleles and MixMHC2pred across 75 Class II alleles. For Class I alleles, positive pairs were selected per allele and peptide length by retaining peptides with $\leq 1\%$ rank and had top 1% scores with a minimum positive score threshold among all scored protein fragments, and negative pairs were randomly sampled from the bottom 50% of MixMHCpred scores per allele to match the distribution of positive pairs. Similarly, for Class II alleles, positive pairs were those ranked in the top 1% per allele and peptide length, and negative pairs were generated using weak-binding thresholds and those at the bottom 1% were retained. For Class II alleles, MixMHC2pred's predicted peptide core was used, and samples were deduplicated based on the core. All the selected synthetic pairs were filtered to exclude any peptide-allele combinations present in the experimental datasets.

### D.3. TCR Data Processing

Each paired T cell receptor (TCR) alpha and beta chain was processed independently using Stitchr (Heather et al., 2022) and ANARCI (Dunbar & Deane, 2016). Only TCR pairs with complete V gene, CDR3 sequence, and J gene information available for both chains were retained for processing. For each chain, Stitchr generated the full-length TCR sequence from the input V gene, CDR3 sequence, and J gene calls. ANARCI was then used to annotate the CDR regions and validate the sequence generated by Stitchr. TCR pairs where either chain failed processing through Stitchr, or failed CDR3 regions validation by ANARCI, were excluded from further analysis. In datasets where full-length $\alpha$ and $\beta$ chain sequences were already available (e.g., the OTS dataset), we performed an additional validation to confirm that sequences generated by Stitchr matched the original ones. For VDJdb, where we have the information on the full TCR-pMHC complex, we filtered out the samples with no available HLA allele for them.

### D.4. Unpaired TCR Repertoire Data

We collected and processed TCR sequences from Raybould et al. (2024) for the pre-training dataset. From the dataset, 1,511,895 distinct full-length TCR sequences were obtained after processing through Stitchr and ANARCI.

### D.5. TCR-pMHC Interaction Data

Experimentally validated immunogenic TCR-pMHC pair sequence data was collected from VDJdb (Bagaev et al., 2020), a curated database of T cell receptor sequences with experimentally validated antigen specificities (vdjdb.cdr3.net). We used the vdjdb-2024-11-27 release, downloaded from the official GitHub repository.

Similar to the pre-training dataset, the TCRs from VDJdb were processed through Stitchr and ANARCI. Only sequences that were successfully processed through both tools were retained. TCR entries with missing MHC allele information were excluded from the final dataset. Post-processing we obtained a total of 31,643 TCR sequences for further analysis.

### D.6. Quantitative pMHC binding data

Quantitative binding data were curated from the Immune Epitope Database (IEDB)(Vita et al., 2019) by filtering for peptides ligands to HLA-A, HLA-B, and HLA-C alleles with 9 amino acids in length and removing entries containing non-canonical residues. To stabilize variance across the broad range of binding affinities, $IC_{50}$ values were transformed to a $\log_{10}$ scale.

### D.7. Dataset Summary

Table 6 summarizes the collected data across experimental and synthetic pMHC pairs for both Class I and Class II HLAs, as well as TCR datasets.

*Table 6.* Summary of experimental and synthetic dataset sizes.

| Dataset Type | Category | Count |
|---|---|---|
| **Experimental pMHC (Class I)** | | |
| | Total HLAs | 17,065 |
| | HLAs with validated peptides | 123 |
| | pMHC pairs | 451,828 |
| **Experimental pMHC (Class II)** | | |
| | Total HLAs | 7,718 |
| | HLAs with validated peptides | 81 |
| | pMHC pairs | 548,858 |
| **Synthetic pMHC (Class I)** | | |
| | Positive pairs | 9,464,613 |
| | Negative pairs | 9,464,613 |
| **Synthetic pMHC (Class II)** | | |
| | Positive pairs | 3,623,224 |
| | Negative pairs | 20,432,290 |
| **TCR Datasets** | | |
| | VDJdb | 31,643 |
| | OTS | 1,511,895 |

## D.8. Held-Out Splits

To enable rigorous zero-shot evaluation, we constructed component-specific held-out datasets with no sequence overlap with the corresponding training data.

*Table 7.* Summary of held-out data splits used for zero-shot evaluation.

| Task | Holdout Type | Held-Out Components |
|---|---|---|
| pMHC binding | Allele-level | **Class I:**
A\*66:01, B\*39:05, C\*03:02, B\*27:05, A\*02:52, B\*44:05, A\*34:01, A\*33:01, B\*18:05, C\*06:02
**Class II:**
DQA1\*02:01/DQB1\*03:03,
DPA1\*01:03/DPB1\*06:01,
DRA\*01:01/DRB1\*13:05,
DRA\*01:01/DRB1\*13:03,
DRA\*01:01/DRB1\*04:03,
DRA\*01:01/DRB1\*12:02 |
| TCR–pMHC recognition | Epitope-level | YLQPRTFLL (SARS-CoV-2 Spike), GLCTLVAML (EBV BMLF1) |

For the TCR-pMHC recognition task, we used a dataset generated by Messemaker et al. (2025) that validated a subset of VDJdb entries for the YLQPRTFLL and GLCTLVAML epitopes binding to HLA-A\*02:01, providing experimentally confirmed binders and non-binders.

# E. Ablation Studies

This appendix provides additional ablation experiments supplementing the results in Section 5.

### E.1. Masking Strategy

To isolate the effect of informed masking, we trained on pMHC data alone (without TCR data) comparing uniform masking (15% of all positions) against informed masking (50% peptide, 20% HLA pocket, 10% elsewhere).

*Table 8.* Masking strategy ablation on zero-shot pMHC binding. Mean $\pm$ SD across held-out alleles.

| Model | Class I (10 alleles) | | Class II (6 alleles) | |
|---|---|---|---|---|
| | AUROC | AUPRC | AUROC | AUPRC |
| *Generic pLM* | | | | |
| ESM-2 (650M) | .41±.09 | .43±.05 | .51±.22 | .54±.17 |
| ESM-2 (3B) | .45±.11 | .46±.06 | .56±.18 | .56±.15 |
| *Uniform Masking (pMHC only)* | | | | |
| +pMHC (650M) | .97±.03 | .97±.03 | .87±.04 | .84±.05 |
| +pMHC (3B) | .98±.02 | .98±.02 | .90±.02 | .89±.03 |
| *Informed Masking (pMHC only)* | | | | |
| +pMHC (650M) | .98±.02 | .98±.03 | .88±.06 | .86±.05 |
| +pMHC (3B) | **.98**±.03 | **.98**±.03 | **.92**±.02 | **.91**±.03 |

Informed masking provides consistent improvements, particularly for Class II alleles (+0.02 AUROC with 3B model). This confirms that concentrating supervision on binding-relevant positions provides useful inductive bias for learning MHC-specific binding preferences.

### E.2. Stage Ablation

Table 9 reports the full stage ablation for the epitope-specific TCR recognition task.

*Table 9.* **Epitope-specific TCR recognition.** Given a fixed pMHC, discriminate binding from non-binding TCRs. Both epitopes completely excluded from DecoderTCR training.

| Model | YLQ | | GLC | |
|---|---|---|---|---|
| | AUC | PRC | AUC | PRC |
| *Supervised (Trained on Epitope)* | | | | |
| NetTCR-2.2 | .88 | .82 | .69 | .87 |
| *Zero-Shot Baselines* | | | | |
| ESM-2 (3B) | .36 | .34 | .48 | .71 |
| *DecoderTCR (3B) Ablations* | | | | |
| Stage 1 only | .18 | .29 | .43 | .70 |
| Stage 2 only | .56 | .42 | .46 | .70 |
| Stage 1 & 2 | **.76** | **.70** | **.64** | **.83** |

## F. Generated Sequences

This appendix provides the complete set of designed TCR sequences from the generation experiments in Section 5.5.

We evaluate designed TCR sequences using both structural confidence metrics and sequence-level quality measures.

### F.1. Designed sequences selected for experimental validation

*Table 10.* TCR Designs for YLQ Peptide

| Design_ID | TRAV | TRBV | TRAJ | TRBJ | $CDR3\alpha$ | $CDR3\beta$ | Edit-Dist | iPTM | Method |
|---|---|---|---|---|---|---|---|---|---|
| 30 | TRAV12-1 | TRBV7-8 | TRAJ30 | TRBJ2-2 | CVVNRDDKIIF | CASLDLNTGELFF | 4 | 0.79 | IEGR |
| 29 | TRAV12-1 | TRBV7-8 | TRAJ30 | TRBJ2-2 | CVVNRDDKIIF | CASLDGNTGELFF | 4 | 0.78 | IEGR |
| 15 | TRAV12-1 | TRBV7-9 | TRAJ34 | TRBJ2-2 | CVPYNTDKLIF | CASLEGNTGELFF | 5 | 0.76 | IEGR |
| 16 | TRAV12-1 | TRBV7-9 | TRAJ34 | TRBJ2-2 | CVPYNTDKLIF | CASLDSNTGELFF | 7 | 0.76 | IEGR |
| 17 | TRAV12-1 | TRBV7-9 | TRAJ34 | TRBJ2-2 | CVPYNTDKLIF | CASRLSNTGELFF | 7 | 0.76 | IEGR |
| 19 | TRAV12-1 | TRBV7-9 | TRAJ34 | TRBJ2-2 | CVPYNTDKLIF | CASLVSNTGELFF | 7 | 0.76 | IEGR |
| 21 | TRAV12-1 | TRBV7-9 | TRAJ34 | TRBJ2-2 | CVVYNTDKLIF | CASLVANTGELFF | 6 | 0.76 | IEGR |
| 23 | TRAV12-1 | TRBV7-9 | TRAJ34 | TRBJ2-2 | CVPDNTDKLIF | CASRVANTGELFF | 7 | 0.76 | IEGR |
| 24 | TRAV12-1 | TRBV7-8 | TRAJ43 | TRBJ2-2 | CVVNTFNDMRF | CASLLSNTGELFF | 5 | 0.76 | IEGR |
| 25 | TRAV12-1 | TRBV7-8 | TRAJ43 | TRBJ2-2 | CVVNTFNDMRF | CASLDANTGELFF | 4 | 0.76 | IEGR |
| 37 | TRAV12-1 | TRBV2 | TRAJ47 | TRBJ2-2 | CVVEYGNKLVF | CASLNQNTGELFF | 6 | 0.76 | IEGR |
| 39 | TRAV12-1 | TRBV7-9 | TRAJ34 | TRBJ2-2 | CVPYNTDKLIF | CASLVANTGELFF | 7 | 0.76 | IEGR |
| 40 | TRAV12-1 | TRBV7-9 | TRAJ34 | TRBJ2-2 | CVPYNTDKLIF | CASLDGNTGELFF | 6 | 0.76 | IEGR |
| 41 | TRAV12-1 | TRBV2 | TRAJ43 | TRBJ2-2 | CVVNEFNDMRF | CALTGYNTGELFF | 5 | 0.76 | IEGR |
| 42 | TRAV12-1 | TRBV7-8 | TRAJ43 | TRBJ2-2 | CVVNTFNDMRF | CASLEANTGELFF | 3 | 0.76 | IEGR |
| 43 | TRAV12-1 | TRBV7-8 | TRAJ43 | TRBJ2-2 | CVVNTFNDMRF | CASLEGNTGELFF | 4 | 0.76 | IEGR |
| 44 | TRAV12-1 | TRBV7-8 | TRAJ30 | TRBJ2-2 | CVVNRDDKIIF | CASTALNTGELFF | 5 | 0.76 | IEGR |
| 12 | TRAV12-1 | TRBV19 | TRAJ43 | TRBJ2-2 | CVVNEFNDMRF | CALSTGNTGELFF | 5 | 0.75 | IEGR |
| 13 | TRAV12-1 | TRBV2 | TRAJ43 | TRBJ2-2 | CVVGNNNDMRF | CALSGQNTGELFF | 4 | 0.75 | IEGR |
| 14 | TRAV12-1 | TRBV7-9 | TRAJ34 | TRBJ2-2 | CVPDNTDKLIF | CASLVANTGELFF | 7 | 0.75 | IEGR |
| 3 | TRAV12-1 | TRBV5-1 | TRAJ43 | TRBJ2-2 | CVVNVNNDMRF | CASSRDGAGELFF | 5 | 0.76 | OneShot |
| 4 | TRAV12-1 | TRBV7-9 | TRAJ34 | TRBJ2-2 | CVPYNTDKLIF | CASTLSNTGELFF | 7 | 0.76 | OneShot |
| 5 | TRAV12-1 | TRBV7-8 | TRAJ30 | TRBJ2-2 | CVVNRDDKIIF | CASLDFNTGELFF | 4 | 0.76 | OneShot |
| 6 | TRAV12-1 | TRBV5-1 | TRAJ43 | TRBJ2-2 | CVVNNNNDMRF | CASSRDGAGELFF | 5 | 0.76 | OneShot |
| 8 | TRAV12-1 | TRBV7-9 | TRAJ34 | TRBJ2-2 | CVPYNTDKLIF | CASLEANTGELFF | 6 | 0.76 | OneShot |
| 49 | TRAV12-1 | TRBV7-9 | TRAJ34 | TRBJ2-2 | CVPDNTDKLIF | CASLESNTGELFF | 6 | 0.76 | OneShot |
| 50 | TRAV12-1 | TRBV7-9 | TRAJ34 | TRBJ2-2 | CVPYNTDKLIF | CASLESNTGELFF | 6 | 0.76 | OneShot |
| 48 | TRAV12-1 | TRBV2 | TRAJ47 | TRBJ2-2 | CVVEYGNKLVF | CASLMPNTGELFF | 5 | 0.75 | OneShot |
| 51 | TRAV12-1 | TRBV5-1 | TRAJ43 | TRBJ2-2 | CVVNFNNDMRF | CASSPANTGELFF | 3 | 0.75 | OneShot |
| 52 | TRAV12-1 | TRBV19 | TRAJ43 | TRBJ2-2 | CVVNEFNDMRF | CASSPGNTGELFF | 4 | 0.75 | OneShot |
| 60 | TRAV12-1 | TRBV7-9 | TRAJ34 | TRBJ2-2 | CVVDNTDKLIF | CASGGGGGGEEFF | 7 | 0.77 | Greedy |
| 56 | TRAV12-1 | TRBV2 | TRAJ43 | TRBJ2-2 | CVVNNFNDMRF | CASSGGNTGEEFF | 5 | 0.76 | Greedy |
| 58 | TRAV12-1 | TRBV7-9 | TRAJ34 | TRBJ2-2 | CVVNNTDKLIF | CASLGGGGGELFF | 6 | 0.76 | Greedy |
| 61 | TRAV12-1 | TRBV7-9 | TRAJ34 | TRBJ2-2 | CVVNNTDKLIF | CASSGGGTGELFF | 5 | 0.76 | Greedy |
| 55 | TRAV12-1 | TRBV2 | TRAJ47 | TRBJ2-2 | CVVEYGNKLVF | CASSGGNAGEEFF | 8 | 0.75 | Greedy |

### F.2. Sequence Analysis

**Design diversity and novelty**   All validated binders are novel, with Levenshtein distance $\geq 3$ from the design scaffold CDR3$\alpha\beta$ recognizing YLQPRTFLL in VDJdb (mean $5.8 \pm 1.4$). Since all YLQPRTFLL-binding TCRs were held out during training, this confirms *de novo* design for an unseen epitope rather than memorizing known binders. Selected binders show mean pairwise distance of 4.2, indicating that IEGR explores diverse sequence solutions rather than converging to a single design motif.

## G. Experimental Protocols

This appendix details the wet-lab validation protocols for the experimental results reported in Section 5.5.

### G.1. Cell Lines

CD8$^+$ TCR-knockout NFAT-Luciferase Reporter Jurkat cells (BPS Bioscience #78757), were engineered using CRISPR to knock out *FAS* creating a *TCR-/FAS-* cells line. Cells were cultured in RPMI 1640 medium (Thermo Fisher Scientific #11875093) containing 10% fetal bovine serum and 1% penicillin-streptomycin and maintained in a 37°C incubator with 5% CO2.

### G.2. TCR Vector Design and Synthesis

Designed TCR sequences were individually synthesized and cloned by Twist Biosciences into their pTwist EF1 Alpha mammalian expression vector, with the TCR under control of the human EF1a promoter. Cloned plasmids were prepared by Twist Biosciences and used directly in nucleofection after resuspension to 500ng/uL in Tris-EDTA pH 8.0.

### G.3. TCR Expression/Nuclefection

*CD8+ TCR-/FAS-* Jurkats were nucleofected using Lonza's 4D nucleofector X-Unit in 16-well strip format using SE Cell Line 4D-Nucleofector® X Kit S (# V4XC-1032) and pulse code CL-120. For each TCR-expressing plasmid, 200,000 cells each and 650ng TCR plasmid in 20 $\mu$L SE combined + Supplement 1 (4.5:1), were pulsed, and after a 10 minute recovery in 80 $\mu$L RPMI, plated in 400 $\mu$L pre-warmed complete media. Cells were incubated as above for 48 hours, followed by dextramer staining and readout.

### G.4. Dextramer Staining and Flow Cytometry

Each full well of nucleofected cells was recovered after 48 hours and washed once with FACS Buffer (10% fetal bovine serum and 2mM EDTA in 1x phosphate buffered saline). Cells were then stained for 10 minutes at room temperature with 10 $\mu$L of HLA-A*02:01/YLQPRTFLL-APC dextramer (Immudex # WB05824). Cells were washed again, then stained with 1.2 $\mu$L Brilliant Violet 421™ anti-human CD3 Antibody (Biolegend #300434) for 20 minutes at room temperature. Three more washes of 200 $\mu$L were performed, followed by staining with 0.2 $\mu$g/mL Propidium Iodide (Thermo Fisher Scientific #P3566) for live/dead discrimination, and analysis on a SONY MA900 Cell Sorter. CD3+/APC+/PI- events exceeding non-binding control background were considered positive.

## H. Additional Results

This appendix provides extended results supplementing the main evaluation in Section 5.

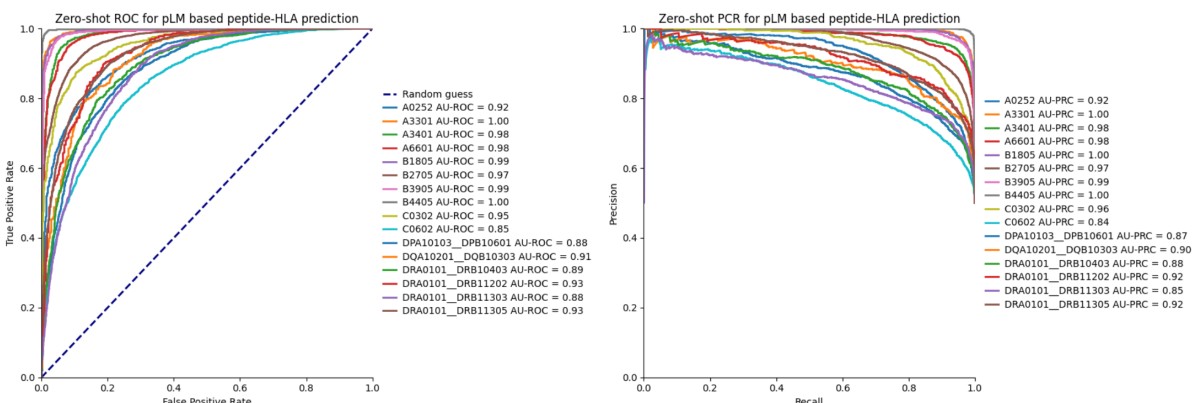

*Figure 5.* **Zero-shot pMHC binding predictions by allele.** Per-allele ROC (left) and PR (right) curves for Stage 1 DecoderTCR model on held-out HLA alleles. All 10 Class I alleles and 6 Class II alleles achieve AUROC $\geq$ 0.85, demonstrating consistent zero-shot generalization across diverse HLA types. Table 1 reports aggregate statistics.

*Table 11.* **Expanded *in silico* evaluation metrics for YLQPRTFLL/HLA-A*02:01 designs.** Five orthogonal metrics across decoding strategies (IEGR, Greedy, OneShot), the structure-based baseline RFdiffusion + ProteinMPNN (PDB 7N6E template, 100 backbones $\times$ 10 sequences = 1,000 designs), and validated positive/negative YLQ controls from Messemaker et al. (2025). IEGR leads on antigen-specific salt-bridge enrichment and sequence naturalness. RFD+MPNN, despite access to a solved crystal structure that DecoderTCR does not require, collapses on OLGA log $P_{gen}$ ($-18.03$ vs. IEGR's $-8.52$, nearly 10 log-units outside the natural repertoire) because ProteinMPNN has no V(D)J prior. Arrows indicate desired direction. Means $\pm$ SD.

| Metric | Pos. YLQ | IEGR | RFD+MPNN | Greedy | OneShot | Neg. YLQ |
|---|---|---|---|---|---|---|
| AF3 ipTM ($\uparrow$) | .733$\pm$.052 | **.644$\pm$.092** | .599$\pm$.084 | .598$\pm$.089 | .612$\pm$.100 | .609$\pm$.095 |
| Boltz-2 ipTM ($\uparrow$) | .847$\pm$.017 | **.843$\pm$.015** | .840$\pm$.020 | .844$\pm$.018 | .840$\pm$.018 | .789$\pm$.054 |
| OLGA log $P_{gen}$ ($\uparrow$) | $-8.79\pm1.31$ | **$-8.52\pm1.59$** | $-18.03\pm1.26$ | $-8.07\pm1.16$ | $-10.61\pm2.86$ | $-9.16\pm1.75$ |
| Salt-bridge ($\uparrow$) | 64.1% | **34.2%** | 28.2% | 17.2% | 21.9% | 37.8% |
| $k$-mer shift ($\downarrow$) | N/A | **.610$\pm$.064** | .790 | .696$\pm$.057 | .629$\pm$.064 | .681$\pm$.050 |

*Table 12.* ***In silico*** **evaluation metrics for GLCTLVAML/HLA-A\*02:01 designs.** Parallel to Table 11 for YLQPRTFLL, three orthogonal metrics evaluated across decoding strategies (IEGR, Greedy, OneShot) and validated positive/negative GLC controls from Messemaker et al. (2025). IEGR leads all generation strategies on every metric, with AF3 ipTM (.715) approaching the positive control (.762) and OLGA $\log P_{\text{gen}}$ ($-8.97$ for positives vs. $-11.35$ for IEGR) substantially closer to the natural repertoire than Greedy ($-12.93$) or OneShot ($-14.78$). Arrows indicate desired direction. Means $\pm$ SD.

| Metric | Pos. GLC | IEGR | Greedy | OneShot | Neg. GLC |
|---|---|---|---|---|---|
| AF3 ipTM ($\uparrow$) | .762±.012 | **.715±.066** | .605±.142 | .637±.091 | .627±.101 |
| OLGA $\log P_{\text{gen}}$ ($\uparrow$) | $-8.97$±1.26 | **$-11.35$±2.75** | $-12.93$±3.52 | $-14.78$±2.16 | $-13.87$±5.47 |
| $k$-mer shift ($\downarrow$) | N/A | **.764±.042** | .799±.030 | .794±.043 | .713±.021 |

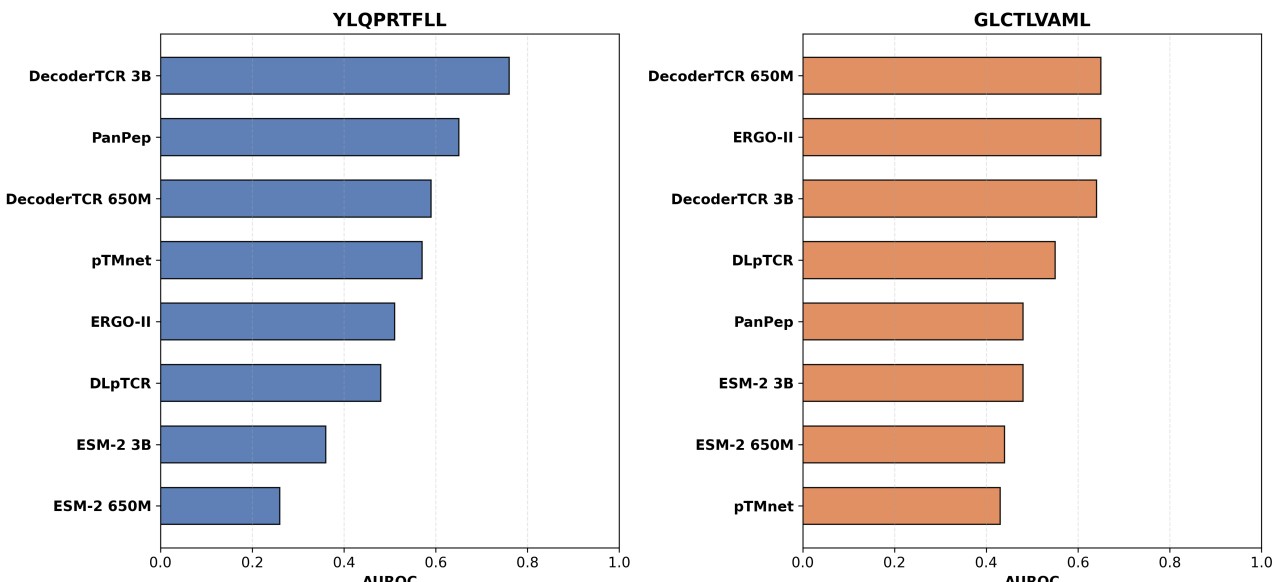

*Figure 6.* **Extended TCR-pMHC recognition comparison.** AUROC for discriminating binding from non-binding TCRs on YLQPRTFLL (left) and GLCTLVAML (right) epitopes. DecoderTCR is compared against other supervised methods (ERGO-II, pMTnet, DLpTCR, PanPep) and generic protein language models (ESM-2 650M/3B). Despite zero-shot evaluation, DecoderTCR approaches or exceeds supervised baselines on both epitopes. Validation data from Messemaker et al. (2025).

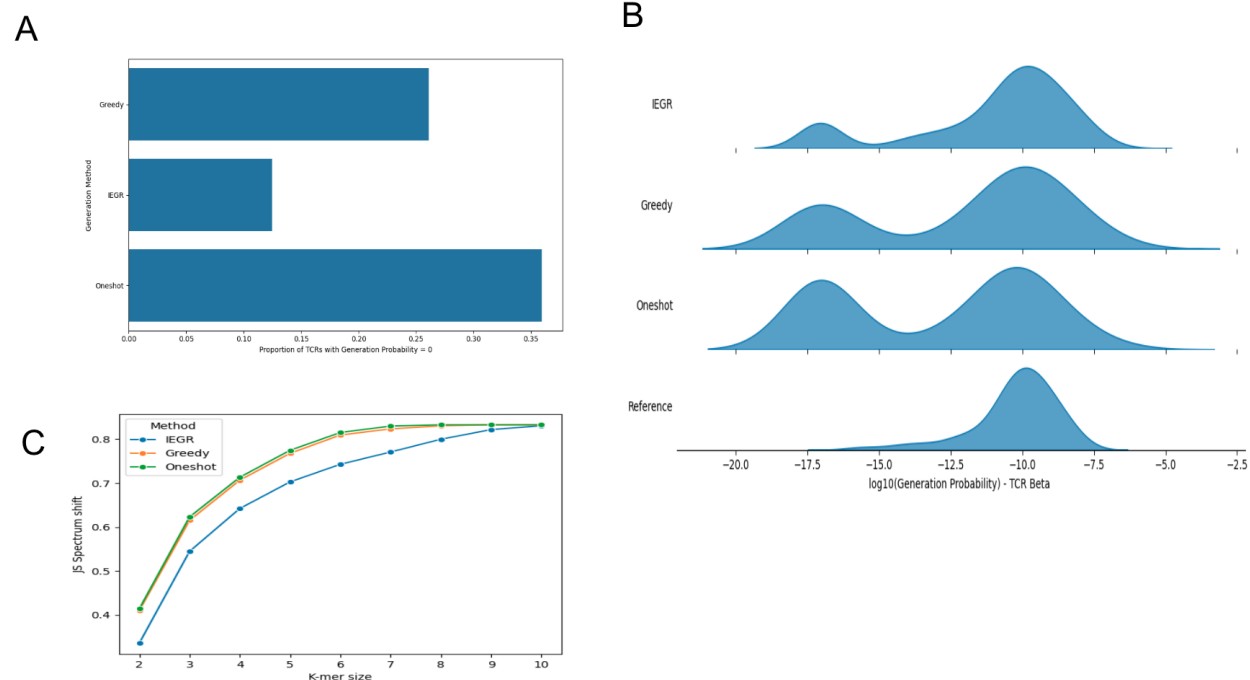

*Figure 7.* **Sequence-level quality of generated TCRs.** Comparison of generation methods targeting YLQPRTFLL/HLA-A*02:01. **(A)** Proportion of generated CDR3$\beta$ sequences with zero V(D)J recombination probability (lower is better) obtained using the tool OLGA (Sethna et al., 2019) **(B)** Distribution of log generation probability across methods; reference distribution shown for comparison. **(C)** K-mer spectrum shift (Jensen-Shannon divergence) relative to reference sequences across k-mer sizes (lower is better). IEGR produces sequences with generation statistics closest to natural TCR repertoires. K-mer spectrum shift calculated as outlined in (Sarkar et al., 2024)

## I. Cross-Paradigm Comparison with Structure-Based and Sequence-Based TCR Design

This appendix supplements the TCR Design discussion in Section 2 by contrasting DecoderTCR with representative sequence-based and structure-based generative pipelines. Table 13 summarizes input modality, data requirements, multi-component conditioning, and zero-shot capability. A quantitative head-to-head comparison against RFdiffusion + ProteinMPNN on YLQPRTFLL/HLA-A*02:01 is reported in Section 5.5.

*Table 13.* **Qualitative comparison of TCR design paradigms.** "Template" refers to a solved or modelled 3D structure of the target complex. "Multi-component" denotes joint modelling of TCR $\alpha/\beta$ chains with peptide and MHC. "Zero-shot" refers to design for antigens unseen during training without target-specific fine-tuning. Method citations appear in Section 2.

| Method | Input | Template | Multi-Component | Decoding | Zero-Shot |
|---|---|---|---|---|---|
| *Sequence-based* | | | | | |
| ERTransformer | Sequence | No | $\beta$ chain only | Autoregressive | Limited |
| TCR-TRANSLATE | Sequence | No | Peptide $\rightarrow$ CDR3 | Autoregressive | Limited |
| **DecoderTCR (ours)** | Sequence | No | Full ($\alpha\beta$ + pMHC) | Non-AR + Gibbs | Yes |
| *Structure-based* | | | | | |
| RFdiffusion + ProteinMPNN | Structure | Yes | Backbone-defined | Diffusion + AR | Limited |
| ADAPT | Structure | Yes | Backbone-defined | Conformational | 0/3 template-free |
| BoltzGen | Structure | Yes | General binder | Diffusion | Limited |

The key practical implication is data regime. Structure-based pipelines require resolved or high-quality predicted templates, which exist for fewer than 500 TCR-pMHC ternary complexes in the PDB and are largely absent for novel disease antigens. ADAPT, the current state of the art for structure-based TCR design, fails to identify binders on template-free targets (Motmaen et al., 2025) and concurrent assessments report that structure prediction methods generalize poorly to

TCR-pMHC interfaces (Ascunce-París et al., 2025; Bradley, 2023). DecoderTCR operates in the complementary regime: zero-shot sequence-first design for novel antigens, leveraging abundant unpaired TCR ($\sim 10^7$) and pMHC ($\sim 10^6$) data via compositional pre-training to compensate for the scarcity of paired and structural supervision.

