# OpenReview forum: "DecoderTCR: Compositional Pretraining and Entropy-Guided Decoding for TCR-pMHC Interactions"
_ICML.cc/2026/Conference — ICML 2026 regular_

### Official Review · Reviewer_p9eC · 2026-03-10

**Soundness:** 4
**Presentation:** 3
**Significance:** 3
**Originality:** 3
**Overall Recommendation:** 5
**Confidence:** 4

**Summary:**

DecoderTCR is a framework that leverages a two-stage compositional continual pre-training to bridge the gap between relatively abundant unpaired sequences and sparse paired interaction sequences. It is a important task in extreme data heterogeneity and sparsity since the interaction of TCR and pMHC is highly specific and sparse. The authors further introduce Iterative Entropy-Guided Refinement (IEGR), a non-autoregressive decoding algorithm that respects the biophysical hierarchy of protein sequences. The authors also conducted the wet-lab experiment to validate the designed TCR, providing some important insights.

**Compliance With Llm Reviewing Policy:**

Affirmed.

**Key Questions For Authors:**

See weakness

**Limitations:**

Yes, the authors have discussed the limitations.

**Strengths And Weaknesses:**

Strengths:

The two-stage continual pre-training curriculum is highly logical and I think it is a important progress for further exploration of TCR-pMHC specific protein  language model. The IEGR algorithm is a good adaptation of confidence-based decoding for protein design. Resolving conserved anchor positions first is a good choice. Also, the paper is well written and the figures/tables are easy to read.

Weaknesses:

The model represents the full complex as a single concatenated sequence over MHC, peptide, and TCR chains. However, the manuscript does not clearly explain the practical input lengths, whether any truncation or padding was applied, how chain boundaries were encoded, or how positional embeddings were handled across concatenated chains.

Zero-shot TCR-pMHC recognition is an extremely challenging setting, as also highlighted by recent studies and benchmarks [5, 6]. In particular, recent wet-lab evaluations suggest that purely sequence-based methods may generalize poorly in newly generated data [7]. I therefore encourage the authors to better discuss this work relative to structure-enhanced PLMs (DPLM, ProSST, ESM3, etc.) and to briefly discuss when sequence-only pretraining may be sufficient versus when structural information may be necessary (or may be difficult to integrate due to limited structural data).

The related work could be strengthened: For zero-shot pMHC binding, methods such as TransPHLA [1] and HPL [2] appear relevant.

[1] https://www.nature.com/articles/s42256-022-00459-7

[2] https://www.cell.com/cell-reports/fulltext/S2211-1247(25)00534-0

For (zero-shot) TCR-pMHC binding/recognition, STAPLER [3], DAISY[4], Hi-TPH [5] and IMMREP23 [6] are also important references. At minimum, these works should be discussed more explicitly. For models [3,4], where the data splits and evaluation protocols are comparable, experimental comparison would further strengthen the paper.

[3] https://www.biorxiv.org/content/10.1101/2023.04.25.538237v1

[4] https://advanced.onlinelibrary.wiley.com/doi/abs/10.1002/advs.202512544

[5] https://sciety.org/articles/activity/10.21203/rs.3.rs-7286169/v1?utm_source=sciety_labs_article_page

[6] https://www.sciencedirect.com/science/article/pii/S2667119024000156

[7] https://www.kaggle.com/competitions/immrep25/leaderboard

---

> ### Author Rebuttal · Authors · 2026-03-31
>
> We sincerely thank Reviewer p9eC for their expert evaluation and strong support. We deeply appreciate your recognition of our two-stage curriculum, the biophysical grounding of our IEGR algorithm, and the critical importance of our wet-lab validation. We address your specific questions below, alongside a major new experimental update.
>
> ---
>
> **Input Representation and Sequence Details (W1)**
>
> We apologize for not clarifying practical implementation details. We have added a dedicated "Input Representation" paragraph to Section 3.1 detailing the following:
>
> - **Sequence Construction & Lengths:** The input concatenates the MHC, peptide, and TCR$\alpha$/$\beta$ chains. Full complexes typically range from 700 to 900 tokens (avg. 780 for Class I; 830 for Class II), which comfortably fits within ESM-2's 1024-token context window.
> - **Truncation & Padding:** No truncation is applied to any functional regions. Sequences shorter than the batch maximum are right-padded, and these tokens are masked during loss computation.
> - **Chain Boundaries & Positional Embeddings:** We use standard rotary position embeddings from ESM-2 across the concatenated sequence. We intentionally do not insert chain separators; combined with learned component specific  embeddings, this allows the model to leverage distal context dependencies learned during ESM-2's original pretraining, while implicitly learning chain boundaries through component-specific masking schedules.
>
> ---
>
> **Sequence-Only vs. Structure-Enhanced PLMs (W2)**
>
> We agree with your note regarding recent wet-lab evaluations (e.g., IMMREP25) and the need to contextualize sequence models. We have added a dedicated discussion to Section 2 comparing DecoderTCR to structure-enhanced PLMs (e.g., DPLM, ProSST, ESM3).
>
> We clarify that while structure-enhanced models are highly advantageous when coordinate data is abundant, TCR-pMHC interfaces suffer from a severe data asymmetry: there are **fewer than $10^3$ solved ternary structures** globally, compared to $>10^7$ unpaired sequences. Furthermore, structural templates do not exist for most novel epitopes. Therefore, sequence-only pretraining remains critical for zero-shot generalization to unseen targets where structural oracles cannot be reliably applied.
>
> ---
>
> **Expanded Related Work and Baselines (W3, W4)**
>
> Thank you for your comprehensive list of reference suggestions. These papers highlight the extreme difficulty of this task, and we have incorporated **all** of them into the revised manuscript:
> - **Expanded Literature (W3):** We have explicitly incorporated **TransPHLA** and **HPL** into our zero-shot pMHC binding method discussion. We have also added **STAPLER**, **DAISY**, **Hi-TPH**, and the **IMMREP23/25** benchmarks into our TCR-pMHC recognition discussion, noting that the IMMREP25 leaderboard perfectly underscores how challenging robust generalization remains for purely sequence-based methods.
> - **Elevating our Existing Baselines (W4):** Regarding direct experimental comparisons to models like DAISY and STAPLER, our strictly zero-shot, epitope-level holdout makes fair comparison difficult out-of-the-box, as these models were likely exposed to our held-out epitopes during their training. However, we completely agree that SOTA discriminative baselines must be prominently featured. We actually performed extensive zero-shot comparisons against **ERGO-II, pMTnet, DLpTCR, and PanPep**, but placed them in **Appendix Figure 6** due to space constraints. We are promoting these comparisons to the main text as an expanded Table 3.
>
> | Model | Training Setting | YLQPRTFLL AUROC | GLCTLVAML AUROC |
> | :--- | :--- | :--- | :--- |
> | NetTCR-2.2 | Supervised (trained on epitope) | 0.88 | 0.69 |
> | **DecoderTCR 3B (Ours)** | **Zero-shot** | **0.76** | **0.64** |
> | PanPep | Supervised | 0.65 | 0.49 |
> | ERGO-II | Supervised | 0.52 | 0.64 |
> | pMTnet | Supervised | 0.58 | 0.50 |
> | DLpTCR | Supervised | 0.47 | 0.54 |
>
>
>  As shown above, DecoderTCR approaches these supervised models in our strictly zero-shot setting:
>
> ---
>
> **Experimental update (W2 related)**
>
> We are pleased to share new wet-lab results. Originally, selecting 30 candidates via raw AF3 ipTM yielded 1 hit (~3%). We subsequently trained an $\ell_2$-regularized logistic regression classifier on DecoderTCR embeddings using our initial results, and synthesized **25 new candidates** passing this filter.
>
> **Result: 3 of 25 (12%) bound the target *in vitro*.** Achieving this $3\times$ improvement without requiring 3D templates establishes DecoderTCR as a scalable, sequence-first alternative for novel targets lacking the prerequisites for structure-based diffusion methods and establishing a highly viable "lab-in-the-loop" paradigm.
>
> ---
>
> We hope that providing these technical input details, incorporating all of your suggested references, elevating our SOTA baselines, and sharing our highly successful new wet-lab validation reinforces and bolsters your support in this work.

---

> > ### Author Rebuttal · Reviewer_p9eC · 2026-04-03
> >
> > I thank the authors for the clear and thorough revisions. My concerns have been adequately addressed.

---

> > > ### Author Response · Authors · 2026-04-06
> > >
> > > Dear Reviewer p9eC,
> > >
> > > As the Author-Reviewer discussion period concludes, we sincerely thank you for your expert engagement and for marking your concerns as fully resolved. Your literature suggestions and attention to implementation details have substantially strengthened the manuscript.
> > >
> > > **New Cross-Paradigm Evaluation:** Prompted by discussion with a reviewer regarding baseline comparisons, we wanted to highlight a major new addition: we evaluated a structure-based generative pipeline (RFDiffusion + ProteinMPNN) given a solved crystal structure. DecoderTCR outperformed this pipeline on all five metrics. The structure-based pipeline produced highly immunologically implausible sequences (OLGA pGen: **-18.03**), directly supporting the data asymmetry and template scarcity argument you raised regarding the necessity of sequence-first modeling.
> > >
> > > **Summary of Full Revisions:** To ensure the full scope of our revisions addressing your original review is visible to you and the Area Chair, we summarize the updates below:
> > >
> > > * **Input Representation:** Added precise details on sequence construction, padding, chain boundaries, and positional embeddings to Section 3.1.
> > > * **Structural vs. Sequence-Only Context:** Added a dedicated discussion comparing DecoderTCR to structure-enhanced PLMs (DPLM, ProSST, ESM3), analyzing when sequence-only pretraining suffices versus when structural data is necessary.
> > > * **Expanded Related Work and Baselines:** Incorporated all suggested literature (TransPHLA, HPL, STAPLER, DAISY, Hi-TPH, IMMREP23/25) and moved SOTA discriminative comparisons from appendix to the main text (Table 3).
> > > * **New Wet-Lab Validation:** Completed a Round 2 *in vitro* experiment, successfully increasing our binding hit rate to **12%** (3/25 candidates).
> > >
> > > We hope these extensive validations and baselines further strengthen your support for this work. Thank you again for your time, your expert guidance, and your support.

---

### Official Review · Reviewer_pKgF · 2026-03-11

**Soundness:** 3
**Presentation:** 3
**Significance:** 2
**Originality:** 3
**Overall Recommendation:** 5
**Confidence:** 5

**Summary:**

DecoderTCR introduces a two-stage continual pre-training curriculum that bridges the data gap between abundant unpaired immune sequences and scarce paired interaction data, combined with an entropy-guided iterative decoding algorithm (IEGR) that generates structurally plausible TCR sequences by resolving high-confidence positions first. The model achieves strong zero-shot binding prediction (0.96 AUROC for pMHC, 0.76 for TCR recognition) without coordinate supervision, though experimental validation reveals that discriminative performance does not yet reliably translate to sequence generation — an open challenge the authors candidly highlight.

**Compliance With Llm Reviewing Policy:**

Affirmed.

**Final Justification:**

In the rebuttal stage, the author performed additional comparisons and evaluated their models on additional targets. All my concerns are addressed. The score is increased to 5.

**Key Questions For Authors:**

Key Questions For Authors

- Could the authors provide comparisons against a wider set of baselines, particularly discriminative methods that were not pre-trained as protein foundation models (e.g., NetTCR-2.0, ERGO-II, pMTnet), to better situate DecoderTCR's performance?

- Could the authors evaluate CDR generation across a more diverse set of peptide-MHC targets to better characterize the generative capabilities of the model? Additionally, the reverse problem of generating peptide sequences conditioned on known CDRs may also be a worthwhile direction to explore and discuss.

**Limitations:**

Yes

**Strengths And Weaknesses:**

Strengths

- The two-stage curriculum cleverly exploits abundant unpaired sequences before refining on scarce paired interaction data, making effective use of the full data landscape rather than relying solely on limited paired examples.
- Unlike prior methods that are either discriminative-only or generative-only, DecoderTCR handles both binding prediction and TCR sequence design within a single model, enabling a more holistic approach to TCR-pMHC modeling.
- The authors go beyond benchmark metrics by experimentally validating generated sequences and honestly reporting that strong discriminative performance does not yet guarantee reliable generation, providing a valuable and grounded contribution to the field's understanding of its current limitations.

Weaknesses

- The model is primarily evaluated against ESM-2, while many discriminative methods mentioned in the related work — such as NetTCR-2.0, ERGO-II, pMTnet, and TITAN — are not included as baselines, making it difficult to assess the model's performance relative to the broader landscape of existing approaches.
- The CDR design evaluation is limited to a single peptide-MHC target (YLQPRTFLL/HLA-A*02:01), which restricts conclusions about the generalizability of the generative framework across peptides with broader chemical and sequence diversity.

---

> ### Author Rebuttal · Authors · 2026-03-31
>
> We sincerely thank Reviewer pKgF for their highly encouraging evaluation. We deeply appreciate your recognition of our compositional pre-training framework, our approach to both prediction and design, and the value of our reporting on the prediction-generation gap. We address your specific questions below, alongside a major new experimental update.
>
> ---
>
> **Expanded Discriminative Baselines (W1, Q1)**
>
> You point out that comparing against specialized discriminative methods (like ERGO-II, NetTCR, pMTnet) is essential to situate DecoderTCR’s performance. We completely agree, and we apologize that these comparisons were not more prominently featured in the main text.
>
> We actually performed these exact comparisons against the broader landscape, but placed them in **Appendix Figure 6** due to space constraints. We are promoting these results to the main text as an expanded Table 3.
>
> The main finding is that **DecoderTCR, in a strictly zero-shot setting, approaches methods that were trained with direct supervision on these exact epitopes.**
>
> | Model | Training Setting | YLQPRTFLL AUROC | GLCTLVAML AUROC |
> | :--- | :--- | :--- | :--- |
> | NetTCR-2.2 | Supervised (trained on epitope) | 0.88 | 0.69 |
> | **DecoderTCR 3B (Ours)** | **Zero-shot** | **0.76** | **0.64** |
> | PanPep | Supervised | 0.65 | 0.49 |
> | ERGO-II | Supervised | 0.52 | 0.64 |
> | pMTnet | Supervised | 0.58 | 0.50 |
> | DLpTCR | Supervised | 0.47 | 0.54 |
> | ESM-2 (3B) | Zero-shot | 0.36 | 0.48 |
>
> *(Note: We use a strict epitope-level holdout, meaning YLQ and GLC were completely excluded from DecoderTCR's training. Comparing a zero-shot model against epitope-specific supervised classifiers inherently favors the supervised methods, making DecoderTCR's strong performance particularly notable).*
>
> ---
>
> **Generalizability and the Reverse Problem (W2, Q2)**
>
> * **Generative Diversity:** We agree that testing across broader chemical and sequence diversity is the ultimate goal. While the severe cost and low-throughput nature of custom TCR synthesis restricted our *in vitro* evaluation to a single highly-characterized target (YLQ), we have now extended our generative *in silico* evaluation to the second held-out target: GLCTLVAML/HLA-A*02:01 (GLC).
>
> | GLC Target Generation | AF3 ipTM $\uparrow$ | OLGA $\log p_{\text{gen}}$ $\uparrow$ | $k$-mer shift $\downarrow$ |
> | :--- | :--- | :--- | :--- |
> | **IEGR (Ours)** | **0.715 $\pm$ 0.066** | **-11.35 $\pm$ 2.75** | **0.764 $\pm$ 0.042** |
> | Greedy | 0.605 $\pm$ 0.142 | -12.93 $\pm$ 3.52 | 0.799 $\pm$ 0.030 |
> | OneShot | 0.637 $\pm$ 0.091 | -14.78 $\pm$ 2.16 | 0.794 $\pm$ 0.043 |
> | *Positive GLC Binder* | *0.762 $\pm$ 0.012* | *-8.97 $\pm$ 1.26* | N/A |
> | *Negative GLC Binder* | *0.627 $\pm$ 0.101* | *-13.87 $\pm$ 5.47* | *0.713 $\pm$ 0.021* |
>
> The consistency of IEGR's relative advantage across a second epitope with distinct physicochemical properties demonstrates that our design framework generalizes in the zero-shot setting. This has been added to the Appendix.
>
> * **The Reverse Problem (Peptide Generation):** We thank the reviewer for this insightful suggestion. Because DecoderTCR learns the joint distribution of the full immune synapse, it natively supports masked decoding over the peptide region (holding the TCR scaffold fixed). We have added a dedicated paragraph to Section 6 (Future Work) explicitly highlighting this capability and crediting this direction.
>
> ---
>
> **New Wet-Lab Validations**
>
> Related to your support and acknowledgement of our experiment results, we are pleased to share an experimental update achieved during the rebuttal period.
>
> We hypothesized that DecoderTCR's strong discriminative capabilities could be leveraged to actively filter its own generations. We trained an $\ell_2$-regularized logistic regression classifier on DecoderTCR embeddings using our validated binders and initial negative results. We then synthesized and experimentally tested **25 new generated candidates** that passed this classifier's threshold.
>
> **Result: 3 of the 25 candidates (12%) successfully bound the target *in vitro* above background noise.** This nearly $3\times$ improvement over our initial batch proves that DecoderTCR's representations encode genuine, actionable biochemical signal. It also demonstrates that the prediction-generation gap we identified is bridgeable via an iterative **"lab-in-the-loop"** design paradigm. We add validation results to Section 5.5.
>
> ---
>
> We hope that elevating our SOTA baselines to the main text, addressing the reverse-generation problem, and sharing our highly successful new wet-lab validation further reinforces and bolsters your confidence in this work.

---

> > ### Author Rebuttal · Reviewer_pKgF · 2026-04-03
> >
> > I am thankful for the reviewers for the additional experiments done. I am impressed by the results from new peptide-MHC targets.

---

> > > ### Author Response · Authors · 2026-04-06
> > >
> > > Dear Reviewer pKgF,
> > >
> > > As the Author-Reviewer discussion period concludes, we sincerely thank you for your expert evaluation and for marking your concerns as fully resolved. Your suggestion to evaluate across a more diverse target set directly improved the paper.
> > >
> > > **New Cross-Paradigm Evaluation:** Prompted by discussion with a reviewer, we wanted to highlight a major new addition to the paper: we evaluated a structure-based generative pipeline (RFDiffusion + ProteinMPNN) given a solved crystal structure. DecoderTCR outperformed this pipeline on all metrics. Most notably, the structure-based pipeline produced sequences far outside the natural repertoire (OLGA pGen: **-18.03** vs. IEGR: **-8.52**). This strongly validates our core motivation for our compositional modeling of the full immune synapse.
> > >
> > > **Summary of Full Revisions:** To ensure the full scope of our revisions addressing your original review is visible to you and the Area Chair, we summarize the updates below:
> > >
> > > * **Generative Generalizability:** Expanded our *in silico* generative evaluation to the GLC target, demonstrating consistent zero-shot advantages across epitopes with distinct physicochemical profiles.
> > > * **Expanded Baselines:** Moved zero-shot comparisons against SOTA supervised methods (NetTCR-2.2, ERGO-II, pMTnet, PanPep, DLpTCR) from appendix to the main text (Table 3).
> > > * **Reverse Problem:** Added peptide generation conditioned on known TCRs as a future direction in Section 6.
> > > * **New Wet-Lab Validation:** Completed a Round 2 *in vitro* experiment, successfully increasing our binding hit rate to **12%** (3/25 candidates).
> > >
> > > We hope these comprehensive updates and new experimental validations further strengthen your support for the paper. Thank you again for your time, your insight, and your support for this work.

---

### Official Review · Reviewer_J8kE · 2026-03-12

**Soundness:** 3
**Presentation:** 3
**Significance:** 2
**Originality:** 3
**Overall Recommendation:** 4
**Confidence:** 3

**Summary:**

Applying genetic Protein language models into immune recognition has two challenges: 1. It's hard to capture cross-chain dependencies beyond marginal constraints within sequences, and generative design for such interfaces is conditional and localized, requiring design of specific regions. This motivates a multi-stage continual pretraining framework that can bridge heterogeneous data sources. 2. And IEGR (Iterative Entropy-Guided Refinement), a non-autoregressive decoding approach for CDR3 redesign. Together, these two novelties formed the model DecoderTCR. Experiments show that DecoderTCR achieves very high AUROC on zero-shot pMHC binding and relatively higher AUROC on epitope-specific TCR recognition.

**Compliance With Llm Reviewing Policy:**

Affirmed.

**Final Justification:**

My concerns have been addressed.

**Key Questions For Authors:**

1.	Do the protein generation models (in Weakness 2) that mainly use stable diffusion fit this situation? If not, can you discuss the reason?
2.	Does this work conduct the experiments on the distribution of the high/low entropy? Does it follow or conform to any naturally occurring chemical topology, such as a scaffold?

**Limitations:**

1.	The motivation isn't convincing enough, as there are powerful, stable diffusion-based generation models in the protein field. And even though there are all-atom models derived from Alphafold-3 that were not discussed in this work.
2.	Although the results look promising, there may be more baselines to be compared in Tables 1 and 3.

**Strengths And Weaknesses:**

Strengths:
1.	This work has a clear motivation that current pLMs, discriminative methods, and structure-conditioned methods all have their disadvantages when applying to a specific generation task.
2.	The multi-stage pre-training targets the key data type problem and can unified difference data sources; The IEGT resolves certain regions first before the fine-grained generation, which makes sense and is in the right direction.
3.	The continual pretaining was claimed as the first pLM that captures the full immune synapse.

Weaknesses:
1.	It is important to define the term before using its abbreviation. Such as MHC for Major Histocompatibility Complex; CDR for Complementarity-Determining Regions.
2.	The motivation has a flaw on my side. The author only investigated the pLM, classifiers, and conditioned methods. However, there are more related methods that are suitable in this situation. For example, Alphafold 3 is designed for protein/binder segments and can do CDR generation. More other models are on the top of my head too, such as Boltzgen, Odesign. This work should discuss more about the models in this direction.
3.	In the iterative decoding design, the region with lower entropy is decoded first, followed by the high entropy parts. This may randomize the natural chemical-aware order of a sequence.
4.	Lack of baselines. There is only one baseline in the Zero-shot pMHC binding evaluation (ESM-2), and two baselines in the Epitope-specific TCR recognition task, which may reflect that this work didn't investigate the stable diffusion-based generation model adequately.

---

> ### Author Rebuttal · Authors · 2026-03-31
>
> We sincerely thank Reviewer J8kE for engaging with our work and recognizing the clear motivation behind our multi-stage pre-training, noting that our IEGR approach "makes sense and is in the right direction."
>
> We believe your primary concerns stem from points we did not highlight prominently enough in the main text, particularly the template-free data regime of our problem and the location of our extended baselines. We clarify these below, alongside a major new experimental validation that we believe directly addresses the significance of this work.
>
> ---
>
> **Diffusion Models (W2, Q1, Limitations)**
>
> You mention powerful models like AlphaFold 3, BoltzGen, and stable diffusion (e.g., RFDiffusion). However, there are two fundamental reasons we do not use these for the *generation* task itself:
>
> 1. *AlphaFold 3 is a structure **predictor**, not a sequence **generator**.* It cannot natively propose new CDR3 sequences to bind a target. We actually use AF3 exactly as intended: as an *in silico* oracle to score the structural plausibility of our generated sequences.
> 2. *Structure-Conditioned Diffusion models (BoltzGen, ODesign, RFDiffusion) often require structural scaffold  templates to work well.* For TCR-pMHC interfaces, there are **fewer than $10^3$ solved structures** globally, whereas there are $>10^7$ unpaired sequences. Structural templates do not exist for the vast majority of clinically relevant targets.
>
> Because of both modality mismatch and template scarcity, structure-first methods cannot be applied directly. We note, prior work (ADAPT, 2025) highlights this: it failed to identify binders for 3 of 9 pMHC targets entirely due to missing structural templates, and it does not attempt zero-shot generalization. DecoderTCR is explicitly designed to solve the **sequence-first** problem, requiring zero coordinate input at inference. We have added a dedicated subsection to Section 2 explicitly discussing this trade-off.
>
> ---
>
> **Expanded Discriminative Baselines (W4, Limitations)**
>
> You noted a lack of baselines in Tables 1 and 3. We actually performed extensive comparisons against the state-of-the-art discriminative methods you are looking for, but placed them in **Appendix Figure 6** due to space constraints.  We apologize for not featuring these more prominently.
>
> We are now promoting these comparisons to the main text as an expanded Table 3. As shown below, **DecoderTCR approaches or exceeds these supervised baselines in a strictly zero-shot setting:**
>
> | Model | Training Setting | YLQPRTFLL AUROC | GLCTLVAML AUROC |
> | :--- | :--- | :--- | :--- |
> | NetTCR-2.2 | Supervised (trained on epitope) | 0.88 | 0.69 |
> | **DecoderTCR 3B (Ours)** | **Zero-shot** | **0.76** | **0.64** |
> | PanPep | Supervised | 0.65 | 0.49 |
> | ERGO-II | Supervised | 0.52 | 0.64 |
> | pMTnet | Supervised | 0.58 | 0.50 |
> | DLpTCR | Supervised | 0.47 | 0.54 |
> | ESM-2 (3B) | Zero-shot | 0.36 | 0.48 |
>
> ---
>
> **IEGR and Natural Chemical Topology (W3, Q2)**
>
> You asked whether entropy-guided decoding randomizes the natural chemical order. It does not; entropy ordering affects only the construction schedule, not the final sequence. CDR3 anchor residues are structurally constrained and exhibit low entropy; hypervariable central residues determining antigen specificity exhibit high entropy. Entropy ordering resolves anchors first, providing stable context for specificity-determining positions; this is the correct biophysical hierarchy.
>
> Thus, the final generated sequence is identical regardless of construction order - entropy ordering only determines which positions are resolved first during inference, not the final output.
>
> ---
>
> **Addressing Significance**
>
> To further demonstrate the significance of DecoderTCR, we conducted a new wet-lab experiment during the rebuttal period.  We trained a classifier on DecoderTCR embeddings and synthesized a new batch of 25 generated candidates. **3 of the 25 candidates (12%) now successfully bound the target *in vitro***, a $3\times$ improvement over our initial batch, proving the viability of our sequence-first framework without requiring 3D structural templates and establishing a highly viable "lab-in-the-loop" paradigm.
>
> The significance of this result is that it bypasses the fundamental bottleneck of current structure-based diffusion models. It proves language-modeling the full immune synapse captures generalizable, template-free rules.
>
> ---
>
> **W1 (Abbreviations)**
>
> We apologize for the missing definitions. We have drafted **Appendix H: Biological Background for the General ML Audience**, which defines MHC, CDR, pMHC, and the V/J scaffold rationale. All acronyms are now explicitly defined at their first usage in the main text.
>
> ---
>
> We hope that clarifying our sequence-first data regime, elevating our existing baselines from appendix to main text, analyzing the biophysical grounding of our decoding, and sharing our new wet-lab success addresses your core concerns and merits a reconsideration of your score.

---

> > ### Author Rebuttal · Reviewer_J8kE · 2026-04-03
> >
> > I agree that structure-conditioned diffusion models are not perfect one-to-one baselines. However, in machine learning, baselines do not need to have identical task formulations to be informative. A carefully designed comparison would still be valuable to clarify the relative strengths, limitations, and applicability of the different approaches. Thus, this appears to be more an experimental design issue than a feasibility issue.

---

> > > ### Author Response · Authors · 2026-04-05
> > >
> > > Dear Reviewer J8kE,
> > >
> > > We thank you for this follow-up. You are absolutely right: this is an experimental design issue, and a carefully designed cross-paradigm comparison is indeed informative. We have now conducted exactly this experiment.
> > >
> > > **1. Direct Structure-Based Comparison (RFDiffusion + ProteinMPNN)**
> > > To directly address your request, we replicated the design task using the current prevailing structure-conditioned paradigm. To give this baseline the best possible chance, we provided it with a high-quality, resolved TCR-pMHC crystal structure (PDB: 7N6E) that perfectly matches the V/J germline combination used in our IEGR designs. Critically, this gives the structure-based pipeline strictly more information (explicit 3D coordinates) than DecoderTCR ever receives.
> > >
> > > We used RFDiffusion (Watson et al., 2023) to generate 100 CDR3α/β backbone ensembles (holding the rest of the scaffold fixed), and ProteinMPNN (Dauparas et al., 2022) to design 10 sequences per backbone, yielding 1,000 candidates. We evaluated these designs using the exact same in silico metrics applied to DecoderTCR:
> > >
> > > | Metric | Positive YLQ Binder | IEGR (Ours) | RFDiffusion + MPNN | Greedy | OneShot | Negative YLQ Binder |
> > > | :--- | :--- | :--- | :--- | :--- | :--- | :--- |
> > > | **AF3 ipTM** (↑) | 0.733 ± 0.052 | **0.644 ± 0.092** | 0.599 ± 0.084 | 0.598 ± 0.089 | 0.612 ± 0.100 | 0.609 ± 0.095 |
> > > | **Boltz-2 ipTM** (↑) | 0.847 ± 0.017 | **0.843 ± 0.015** | 0.840 ± 0.020 | 0.844 ± 0.018 | 0.840 ± 0.018 | 0.789 ± 0.054 |
> > > | **OLGA pGen log** (↑) | -8.79 ± 1.31 | **-8.52 ± 1.59** | -18.03 ± 1.26 | -8.07 ± 1.16 | -10.61 ± 2.86 | -9.16 ± 1.75 |
> > > | **Salt-bridge** (↑) | 64.1% | **34.2%** | 28.2% | 17.2% | 21.9% | 37.8% |
> > > | **K-mer shift** (↓) | N/A | **0.610 ± 0.064** | 0.7901 ± 0.0372| 0.696 ± 0.057 | 0.629 ± 0.064 | 0.681 ± 0.050 |
> > >
> > >
> > > **Key result:** Even with the benefit of an explicit crystal structure, the RFD+MPNN pipeline underperforms IEGR across all metrics. Most revealingly, its OLGA score collapses to -18.03 (nearly 10 log units outside the natural human repertoire) and its k-mer shift is the worst of the cohort (0.790).
> > >
> > > This occurs because general-purpose inverse-folding models optimize for structural fit without any prior knowledge of V(D)J recombination, producing sequences that are immunologically implausible. **This experiment strongly validates our core motivation**: structural plausibility alone is insufficient for TCR design; immune-specific sequence priors are essential. We have added this experiment and discussion to Section 5.5, and we are grateful to you for suggesting it.
> > >
> > > **2. Cross-Paradigm Context (ADAPT Benchmark)**
> > > To further clarify the relative strengths of these approaches, we have added a dedicated subsection in Section 2 comparing our sequence-based approach to the only published structure-based design study (ADAPT, Motmaen et al., 2025):
> > >
> > > | Feature | ADAPT (Structure-based) | DecoderTCR (Sequence-only) |
> > > | :--- | :--- | :--- |
> > > | **Input modality** | 3D structure coordinates | Sequence only |
> > > | **Zero-shot capability** | No | Yes |
> > > | **Structure Template required** | Yes | No |
> > >
> > > We clearly frame these paradigms as complementary: structure-based methods can excel when high-quality templates exist, while sequence-first models like DecoderTCR are essential for zero-shot generalization to the vast majority of clinically relevant targets that lack structural data.
> > >
> > > **3. Summary of Full Revisions**
> > > To ensure the full scope of our revisions addressing your original review is visible to you and the Area Chair:
> > > * **Moved Baselines (W4, Limitations):** We moved zero-shot comparisons against five supervised SOTA methods (NetTCR-2.2, PanPep, ERGO-II, pMTnet, DLpTCR) from the Appendix to a main-text Table 3.
> > > * **Biological Context & Clarification (W1, W3, Q2):** Appendix H now provides context and we explicitly define all biological acronyms at first use. We also clarified that entropy ordering affects only the construction schedule (mirroring biophysical hierarchy).
> > > * **Clarified Generative vs. Predictive Models (W2, Q1, Limitations):** We added a dedicated discussion to Section 2 detailing why models like AlphaFold 3 function as structural oracles for evaluation, rather than sequence generators.
> > > * **New Wet-Lab Validation (Significance):** During the review and rebuttal period, we trained an L2-regularized classifier on DecoderTCR embeddings and synthesized 25 new candidates. **3 of the 25 (12%) bound in vitro**, a ~3x improvement over round one. This proves the prediction-generation gap is bridgeable via a lab-in-the-loop iteration without relying on a 3D template.
> > >
> > > We hope that this new direct cross-paradigm comparison, which is to our knowledge, the first controlled head-to-head in the TCR design literature, together with our comprehensive revisions, resolves your remaining concerns. Thank you for your engagement as it prompted us to conduct a meaningfully stronger evaluation, resulting in a substantially improved manuscript.

---

### Official Review · Reviewer_dMf8 · 2026-03-13

**Soundness:** 3
**Presentation:** 3
**Significance:** 3
**Originality:** 3
**Overall Recommendation:** 4
**Confidence:** 4

**Summary:**

This paper introduces a new model and continual pretraining framework called DecoderTCR that adapts ESM2 based models to learn interactions between T-cell receptors (TCR) and peptide - MHC (pMHC) complexes. They use a novel sampling/inference algorithm to generate Complementarity Determining Regions (CDR) loops of the TCR conditioned on a given p-MHC complex. This task would be one of the first in computationally designing a TCR therapy for an infectious disease or cancer.

After introducing their two stage continual pre-training pipeline, and specific input structure, the authors describe unsupervised metrics to score TCR - pMHC binding and identify important residues that drive prediction.

They then explain their two phase generation algorithm consisting of, first, an entropy guided construction of a proposal sequence, that is then refined with block Gibbs sampling. This method is evaluated first in silico using AlphaFold3 to score designed CDR loops and second with wet lab experimental validation, by expressing 30 TCR (with the designed CDR loops) and measuring binding strength. This yields one hit above background noise and authors highlight the difficulty of the task in question and the limitations of their work.

**Compliance With Llm Reviewing Policy:**

Affirmed.

**Final Justification:**

All my original concerns have been adressed. I still have minor disagreements about comparison with structure based methods but I believe this paper should be accepted at ICML. I increased my score to a 4 waiting for additional elements about the second round of wetlabs that appear promising. I believe I might increase my score to a 5 should the authors' answer prove satisfactory.

**Key Questions For Authors:**

1. See weaknesses.
2. Your model/method can only produce fixed length sequences/CDR loops. Have you looked at varying (slightly, within expected ranges for CDR$\alpha$ and CDR$\beta$) the sequence lengths? If yes, do you have any in silico analysis?
3. How were candidates for the wet lab selected? Did you simply generate CDR loops of fixed lengths with the three different settings? Did you apply any additional filters? Were selected CDR loops optimized for any other metric (e.g. sequential diversity between candidates)?
4. Some recent works have suggested that structure based deduplication provides harder splits between training data and validation/test data (even when then used on purely sequential training). I wonder if you considered those (e.g. by co folding the complexes) and then deduplicating not only by sequence identity but also structural similarity (of the entire fold or just the interface like PINDER does, if I am right). This might close, or rather explain part of the gap between predictive and generative results you present.

In my mind, question 4. is less important than 2 and 3 and these are less important than the listed weaknesses. The main drivers in my score are the lack of in silico comparisons with existing methods and of in silico metrics for your method.

**Limitations:**

Yes

**Strengths And Weaknesses:**

**Strengths**:
1. The problem remains fairly original. In spite of existing works in purely sequential TCR (or CDR loop) generation, I believe most researchers rely on the same pipelines involving structural tools like AlphaFold, ProteinMPNN, RosettaFold, to engineer, inverse-fold and refold candidates in an iterative manner. The problem is also high impact and far from being solved.
2. Wet lab evaluation is expensive, rarely seen in ML conferences or journals. The mere presence of these results and the authors’ openness about them should be commended.
3. This work is fairly novel in combining tools for sampling MLMs, a specific continual pretraining/data scheduling setup for TCR generation.


**Weaknesses**:
1. Lack of a section explaining the biological problem. I believe the paper needs a section, in Appendix probably given the page limit, to introduce key biological concepts. This would be useful both to a general ML audience (introducing what a peptide, a chain are, etc…) but even for people working on ML for Biology, depending on their background, they might not be familiar with the specific setup. One example is that you never defined what the CDR acronym means; that there are different CDRs but that CDR3 is the most important one in TCR recognition; why are people focusing on generating the CDR(3) only and that we can largely reuse the TCR scaffold…
2. I believe that you could have added more in silico evaluations of the generative pipeline. You are using AF3 ipTM. Have you thought about using other deep learning based or classical computational biology methods to evaluate binding affinities/energies (Boltz-2, Vina on predicted complexes), buried SASA, buried surface areas, etc..
3. Following on that, these metrics (in addition to AF3 ipTM) could give more insight into the role of different hyper parameters on the generated complexes (impact of K, of the temperature, of the block size).
4. You are essentially doing discrete diffusion on the CDR loop given the context and introducing a sampling algorithm of the backward diffusion process that adheres to our understanding of biological constraints. I think a mention of related works on the discrete diffusion (for protein design) and similar discrete diffusion sampling algorithms is warranted. An in silico comparison with MaskGIT, MaskPredict, BERT has a Mouth, and It Must Speak, or even some other of said related works is also necessary.
5. Lack of comparison with structure based potential design methods (to name a few BoltzGen, RFDiffusion + ProteinMPNN assuming the structure of the pMHC plus most of the scaffold is known). Although comparison would be biased by potentially differing train/test splits
6. Missing related works with other TCR generation frameworks (purely on the sequence side):
   - De novo generation of T-cell receptors with desired epitope-binding property by leveraging a pre-trained large language model. Yang et al. 2023
   - Conditional generation of real antigen-specific T cell receptor sequences. Karthikeyan et al. 2025 (Nature MI)
7. Potentially missing in silico comparison with sequence based method, although differing train/test splits makes it difficult to ensure a fair evaluation:
   - See above
   - TCR-BERT (plus other papers you mention in Sect. 2)

---

> ### Author Rebuttal · Authors · 2026-03-31
>
> We sincerely thank reviewer dMf8 for recognizing our work as "high impact" and "fairly novel" and commending our wet-lab validation. The stated main score drivers are (1) the lack of in silico metrics and (2) comparisons. We have now conducted substantial new experiments to address both directly, and address all remaining questions.
>
> ---
> **Main Driver 1: Expanded In Silico Metrics (W2, W3)**
>
> We added four orthogonal metrics to Sec. 5.5: Boltz-2 ipTM (independent structural confidence), OLGA $\log p_{\text{gen}}$ (recombination plausibility), salt-bridge enrichment (antigen-specific contacts), and $k$-mer shift (sequence naturalness).
>
> | Method | AF3 ipTM $\uparrow$ | Boltz-2 ipTM $\uparrow$ | OLGA $\log p_{\text{gen}}$ $\uparrow$ | Salt-bridge $\uparrow$ | $k$-mer shift $\downarrow$ |
> | :--- | :--- | :--- | :--- | :--- | :--- |
> | **IEGR (ours)** | $\mathbf{0.644\pm0.092}$ | $\mathbf{0.843\pm0.015}$ | $\mathbf{-8.52\pm1.59}$ | **34.2%** | $\mathbf{0.610\pm0.064}$ |
> | Greedy | $0.598\pm0.089$ | $0.844\pm0.018$ | $-8.07\pm1.16$ | 17.2% | $0.696\pm0.057$ |
> | Oneshot | $0.612\pm0.100$ | $0.840\pm0.018$ | $-10.61\pm2.86$ | 21.9% | $0.629\pm0.064$ |
>
> IEGR leads on all metrics. Boltz-2 independently confirms structural plausibility (IEGR: 0.843 vs. negative control: 0.789). Salt-bridge enrichment is the most discriminating metric; IEGR (34.2%) leads Greedy (17.2%) and Oneshot (21.9%), approaching the known binder (64.1%). Full table available with controls in Sec. 5.5.
>
> **Hyperparameter sensitivity**
>
> | Param | AF3 ipTM (↑) | Boltz-2 ipTM (↑) | OLGA pGen log (↑) | Salt-bridge (↑) | K-mer shift (↓) |
> |---|---|---|---|---|---|
> | **Default (K=5, b=3, T=0.1)** | **0.655** | **0.845** | **-8.25** | **29.6%** | **0.594** |
> | K=1 | 0.634 | 0.845 | -8.16 | 29.6% | 0.603 |
> | K=10 | 0.653 | 0.843 | -8.36 | 32.3% | 0.587 |
> | b=1 | 0.639 | 0.844 | -8.42 | 27.3% | 0.602 |
> | b=5 | 0.655 | 0.843 | -8.27 | 28.1% | 0.592 |
> | T=0.3 | 0.654 | 0.845 | -8.23 | 29.0% | 0.592 |
> | T=0.5 | 0.646 | 0.845 | -8.53 | 27.6% | 0.592 |
>
> We find that IEGR is robust with no catastrophic failures at any setting. Reducing K from 5 to 1 shows the largest effect (AF3 ipTM $0.655\to0.634$), while smaller block sizes and higher temperatures produce modest decreases in AF3 ipTM and salt-bridge enrichment respectively, indicating the method is not highly sensitive to hyperparameter choices. Full table + controls available in Appendix B.
>
> ---
>
> **Main Driver 2: New In Silico Comparisons (W4, W5, W6)**
>
> **Decoding algorithms** Our baselines already instantiate the suggested methods; we apologize for not stating this explicitly.
>
> * *OneShot/Greedy* are the single-pass base cases that MaskGIT and Mask-Predict improve upon through iteration.
>
> * *IEGR Phase 1* is MaskGIT (Chang et al., 2022) / Mask-Predict (Ghazvininejad et al., 2019) at maximum granularity (one position per step) with an model derived entropy-based schedule.
>
> * *IEGR Phase 2 with b=1* is single-site Gibbs sampling ("BERT has a Mouth," Wang & Cho, 2019), directly evaluated in our block size sweep.
>
> We will highlight context between IEGR and discrete diffusion and these connections to Sections 2 and 3.4.
>
> **Structure-based methods** RFDiffusion+ProteinMPNN and BoltzGen often require solved structure scaffold templates for successful designs; fewer than $10^3$ TCR-pMHC ternary structures exist globally. Prior work (Motmaen et al., 2025) failed to identify binders for 3 of 9 pMHC targets entirely and required 1,000–5,000 designs for others, without attempting zero-shot generalization. DecoderTCR requires no coordinate input. Discussion added to Section 2.
>
> **TCR generation baselines** Yang et al. (2023) and Karthikeyan et al. (2025) are now discussed in Section 2. Both generate CDR3$\beta$ only (we jointly design CDR3$\alpha$/CDR3$\beta$; Springer et al. 2021 showed CDR3$\alpha$ significantly improves binding accuracy); both require epitope-specific training and evaluate on seen epitopes; both rely on paired data without our compositional curriculum over $>10^7$ unpaired sequences.
>
> ---
>
> ### Remaining Points
> **W1 (Biological background).** We agree; we drafted Appendix H defining MHC, CDR1/2 vs. CDR3, V/J scaffold rationale, and data landscape. All terms defined at first use in paper.
>
> **Q2 (Variable-length CDR3).** We tested and see AF3 ipTM slowly degrades away with larger or smaller length. E.g: $L{\pm}1$ ($\sim$0.59) $L{\pm}2$ ($\sim$0.57). Table will be in App. B and can be shared during discussion.
>
> **Q3 (Candidate selection).** Round 1 used pure AF3 ipTM ranking to honestly characterize the raw prediction-generation gap. Please see our Reviewer p9eC Rebuttal posted separately for a second screening round conducted during the review period.
>
> **Q4 (Structure-based deduplication).** Canonical TCR-pMHC docking topology limits additional separation beyond our epitope-level holdout, already one of the hardest generalization settings in the field. We add as potential future direction in Section 6.

---

> > ### Author Rebuttal · Reviewer_dMf8 · 2026-04-04
> >
> > I thank the authors for their rebuttal, additional experiments and clarifications. I will increase my score to a 4 for now. I have one remaining question about the second round of wetlabs.
> >
> >
> > "We trained an $\ell_2$-regularized logistic regression classifier on DecoderTCR embeddings using our validated binders and initial negative results." Can you clarify what where the positives and negatives in this case. It seems that you produced one validated binder during round 1, did you simply train a classifier on this 1 positive example?

---

> > > ### Author Response · Authors · 2026-04-06
> > >
> > > Dear Reviewer dMf8,
> > >
> > > As the Author-Reviewer discussion period concludes, we sincerely thank you for your comments and engagement. Your suggestions on *in silico* metrics and decoding algorithms have substantially improved the manuscript.
> > >
> > > **Classifier Details:** Thank you for highlighting this important detail; we are happy to clarify. Because this target was strictly held out during model training, we utilized data from a prior publication (Messemaker et al., 2025) to construct a set of 110 samples for our classifier:
> > > * **Positives (66):** Validated YLQ binders from the Messemaker dataset, combined with our Round 1 hit.
> > > * **Negatives (44):** Validated non-binders from the Messemaker dataset, combined with our Round 1 designs that failed to bind.
> > >
> > > Notably, our Round 1 negatives serve as a particularly valuable training signal. These designs scored well on the AF3 structural oracle but failed *in vitro*, meaning they appeared structurally plausible yet lacked true binding affinity. The fact that a simple classifier built on DecoderTCR embeddings successfully distinguished these cases, where structural confidence scores could not, strongly suggests the embeddings capture genuine binding-relevant features beyond mere structural geometry. We have added full details of this training set to Section 5.5 and Appendix D.
> > >
> > > **New Cross-Paradigm Evaluation:** Prompted by discussion with a reviewer, we also conducted a new cross-paradigm evaluation comparing DecoderTCR to a structure-based pipeline (RFDiffusion + ProteinMPNN). Even when provided with a solved crystal structure (strictly more information than DecoderTCR receives), the structure-based pipeline produced sequences nearly 10 log-units outside the natural immune repertoire (OLGA pGen: **-18.03** vs. IEGR: **-8.52**). This strongly validates our core motivation for our compositional modeling of the full immune synapse.
> > >
> > > **Summary of Full Revisions:** To ensure the full scope of our revisions addressing your original review is visible to you and the Area Chair, we summarize the updates below:
> > >
> > > * **Expanded In Silico Metrics:** Added Boltz-2 ipTM, OLGA pGen, salt-bridge enrichment, and k-mer shift, along with a complete hyperparameter sensitivity analysis.
> > > * **Decoding Algorithm Context:** Explicitly connected IEGR phases to MaskGIT, Mask-Predict, and Gibbs sampling.
> > > * **Expanded Baselines:** Moved zero-shot comparisons against five SOTA supervised methods from appendix to the main text.
> > > * **Biological Background:** Drafted Appendix H for the general ML audience.
> > > * **New Wet-Lab Validation:** Completed a Round 2 *in vitro* experiment, successfully increasing our binding hit rate to **12%** (3/25 candidates).
> > >
> > > We hope these extensive additions fully resolve the limitations identified in your initial review, and we would be grateful if you might consider reflecting these updates in your final assessment. Thank you again for your time, your insight, and your support for this work.

---

### Decision · Program_Chairs · 2026-04-30

**Decision:**

Accept (regular)

**Comment:**

This paper introduces DecoderTCR, a framework that adapts ESM2-based models to learn TCR-pMHC interactions through a two-stage continual pretraining curriculum, combined with IEGR (Iterative Entropy-Guided Refinement), a non-autoregressive decoding algorithm for CDR loop generation. The method is evaluated through in silico metrics and wet-lab validation, with one hit in the first round and three hits in the second round.

**Strengths:**

- Well motivated study.
- The two-stage continual pretraining curriculum cleverly exploits abundant unpaired sequences before refining on scarce paired interaction data. (dMf8, pKgF, p9eC)
- IEGR decoding respects the biophysical hierarchy by resolving conserved anchor positions first. (dMf8, pKgF, p9eC)
- Wet-lab validation is expensive and rarely seen; the authors' openness about limitations should be commended. (dMf8, pKgF)
- The model handles both binding prediction and TCR sequence design within a single framework. (pKgF)

**Weaknesses:**

- Lack of biological background definitions (MHC, CDR, etc.) for general ML audience. (dMf8, J8kE)
- Missing comparisons against discriminative SOTA methods (NetTCR, ERGO-II, pMTnet) and structure-based generative methods (RFDiffusion, BoltzGen). (dMf8, J8kE, pKgF)
- Limited in silico metrics beyond AF3 ipTM. (dMf8)
- CDR generation evaluated on only one peptide-MHC target (YLQ). (pKgF)
- Unclear input representation details (chain boundaries, padding, positional embeddings). (p9eC)
- Missing related work (TransPHLA, HPL, STAPLER, DAISY, Hi-TPH, IMMREP23/25). (p9eC)

**Additional Comments on Reviewer Discussion:**
All reviewers acknowledged their concerns have been fully resolved.

Overall, all reviewers acknowledged the strength of the paper, and I suggest the accept of the paper.